# Learnable Sampler Distillation for Discrete Diffusion Models

**Feiyang Fu, Tongxian Guo, Zhaoqiang Liu**[*]
University of Electronic Science and Technology of China

## Abstract

Discrete diffusion models (DDMs) have shown powerful generation ability for discrete data modalities like text and molecules. However, their practical application is hindered by inefficient sampling, requiring a large number of sampling steps. Accelerating DDMs by using larger step sizes typically introduces significant problems in generation quality, as it amplifies the impact of both the compounding decoding error due to factorized predictions and discretization error from numerical approximations, leading to a significant decrease in sampling quality. To address these challenges, we propose learnable sampler distillation (LSD), a novel approach to train fast and high-fidelity samplers for DDMs. LSD employs a distillation approach where a student sampler with a few steps learns to align its intermediate score trajectory with that of a high-quality teacher sampler with numerous steps. This alignment is achieved by optimizing learnable sampler coefficients that adaptively adjust sampling dynamics. Additionally, we further propose LSD+, which also learns time schedules that allocate steps non-uniformly. Experiments across text generation, image generation, and synthetic tasks demonstrate that our proposed approaches outperform existing samplers for DDMs, achieving substantially higher sampling quality with significantly fewer sampling steps. Our code is available at https://github.com/feiyangfu/LSD.

## 1 Introduction

Diffusion models have demonstrated remarkable success across various generative tasks, particularly excelling in the synthesis of data within continuous domains like images, audio, and videos [1, 2, 3, 4, 5]. These models frame the data generation process as a gradual denoising procedure in a continuous latent space. However, many other important data modalities, such as natural language, molecular sequences, and categorical data, inherently possess discrete structures. Applying diffusion models directly to these discrete spaces is challenging, as the standard formulation relies on continuous state transitions. Recently, discrete diffusion models (DDMs) [6, 7, 8, 9, 10, 11, 12] have been developed to address this issue. DDMs are specifically designed to operate on discrete data, adapting the core diffusion idea to categorical variables and enabling principled generation. Recent advances in DDMs have shown promising results, achieving competitive performance in generating high-fidelity discrete data. Despite their promising applicability, DDMs face an important challenge in sampling efficiency, and they typically require a substantial number of function evaluations (NFEs), e.g., 1024 or more, making inference computationally expensive.

Current sampling methods for DDMs are mainly divided into two categories: 1) Exact simulation methods [13, 14] provide unbiased samples from the target distribution but suffer from high sampling times and expensive computational costs due to numerous model evaluations, leading to poor scaling with dimensionality. 2) Approximate methods like $\tau$-leaping [15, 16] are designed for parallelization

---

[*]Corresponding author.

39th Conference on Neural Information Processing Systems (NeurIPS 2025).

and potentially faster sampling. However, such methods are first-order accurate and require small step sizes to ensure sampling quality.

Directly accelerating the sampling of DDMs through reducing NFEs typically produces unsatisfactory results, since this amplifies the impact of the compounding decoding error [17] and discretization error. Compounding decoding error arises since DDMs employ a factorized parameterization for computational efficiency, predicting the denoised state of each token independently, and ignoring inherent dependencies between tokens in the sequence. Consequently, the learned factorized denoising distribution differs from the true reversal process. This discrepancy is exacerbated when reducing NFEs, as the approximation quality degrades over larger intervals. Discretization error occurs since large step sizes make it inaccurate for numerical methods like Euler [18] and $\tau$-leaping [15] to approximate the reverse dynamics. Moreover, these two errors accumulate over the sampling trajectory, severely degrading sampling quality when using small NFEs. Throughout the following, we call the combination of compounding decoding error and discretization error as accumulated error for brevity. To address the issue incurred by large accumulated error, we propose learnable sampler distillation (LSD) and its improved version for efficient sampling of DDMs.

## 1.1 Related Work

**Efficient sampling in continuous diffusion models**   Recent efforts to accelerate sampling in continuous diffusion models largely focus on reducing the NFEs for solving the reverse-time ordinary differential equation (ODE) or stochastic differential equation (SDE) [19, 4, 20, 21, 22, 23, 24].

One major direction involves designing advanced ODE solvers. Some works [20, 25, 26, 27, 4] provide efficient sampling methods by establishing high-order numerical ODE solvers for continuous diffusion models.

There are also approaches that learn or optimize various components of the sampling process. AYS [28] seeks non-uniform time step schedules specific to given models and datasets, though their optimization can be computationally intensive. DMN [29] proposes a general framework for designing an optimization problem that seeks more appropriate time steps by minimizing the distance between the ground-truth solution to the ODE and an approximate solution corresponding to the numerical solver. AMED-Solver [30] learns adaptive mean estimation directions based on the observation that trajectories often reside in low-dimensional subspaces, which typically involves training an auxiliary network with high costs.

The most relevant works to us are perhaps LD3 [31] and S4S [32], both proposing learning diffusion model solvers via distillation in the continuous domain. LD3 efficiently learns the time discretization by backpropagating through the ODE-solving procedure using the proposed surrogate loss. S4S further learns the coefficients of the student solver by minimizing the distance between the final samples generated by the student and teacher solvers using learned non-uniform time schedules. However, these approaches face challenges when applied to DDMs. We highlight several distinctions in our learnable sampler distillation (LSD) approach (and its improved version) designed to address these challenges. 1) DDM sampling involves non-differentiable categorical sampling at each step, obstructing direct gradient flow from the final discrete output back to the sampler parameters. The reliance of S4S on final sample comparison is thus infeasible. We address this issue by aligning the intermediate score trajectories between the student and teacher samplers. This provides a viable path for gradient-based optimization of the learnable coefficients within the discrete sampling methods. 2) The work for S4S uses the final sample matching error to learn the time schedules, which may ignore dynamic changes of the accumulated error in the intermediate steps. We instead learn the time steps by aligning the effective transition term at intermediate stages during the reverse process. The effective transition term in the reverse process incorporates step sizes and concrete scores, which are tailored for DDMs. 3) The work for S4S optimizes the continuous initial noise using projected stochastic gradient descent (SGD) within an $L_2$ ball. This is inapplicable to DDMs where the initial state is often a discrete sequence, e.g., all masked tokens, which lacks a continuous gradient. To address this issue, we adapt the approach by measuring proximity using Hamming distance, which is suitable for discrete spaces and does not perform gradient updates on itself.

**Distillation in Diffusion Models**   The distillation of continuous diffusion models is a rapidly advancing field. A prominent direction is related to the consistency model [33], which aims to learn a function that maps any point on an ODE trajectory to its origin, enabling one-step or few-

step generation. This paradigm has been extended to multi-step variants [34, 35] for improved performance. Other significant works focus on directly matching student and teacher distributions, such as distilling guided diffusion models [36], proposing simplified and faster matching objectives [37], recursively distilling a deterministic diffusion sampler into a new model [38], or concentrating on one-step distillation [39]. While these methods are highly effective for continuous models, they usually rely on continuous paths in the sense that the sampling process of each step is differentiable. Our work diverges by proposing a distillation framework specifically for the discrete diffusion model, which does not assume such a continuous path, and addressing a different set of challenges like the non-differentiability of the outputs. A recent work for Di[M]O [40] also involves distilling discrete diffusion models. It distills a multi-step masked diffusion model into a one-step generator. This is achieved by training a new student model from scratch, using a sophisticated proxy objective that involves creating "pseudo-intermediate states" and training an auxiliary model to match conditional output distributions. Our approaches are significantly different with Di[M]O in both goal and mechanism. Similarly to LD3 and S4S, we focus on a few-step sampler distillation. We tackle the challenge of non-differentiability of sampling from categorical distributions, and we enhance an existing sampler rather than replacing the model, which avoids the complexity of training a new generator and an auxiliary model.

**Discrete diffusion models**   DDMs have emerged and undergone substantial development recently. SEDD [6] proposes score entropy, a novel loss that naturally extends score matching to discrete spaces and integrates seamlessly to build DDMs. RADD [9] reveals that the concrete score in absorbing diffusion can be expressed as conditional probabilities of clean data, multiplied by a time-dependent scalar in an analytic form and it unifies absorbing DDMs and any-order autoregressive models.

Various strategies have been proposed to accelerate the sampling of DDMs while maintaining quality. Among these, approximate simulation methods [18, 16, 15, 8] are widely used due to their potential for parallelization. A prominent example is the $\tau$-leaping algorithm [8] that is adapted for DDMs. $\tau$-leaping simulates the process by taking an approximate Euler-like step at each data dimension simultaneously and independently. Tweedie $\tau$-leaping [6, 41] is an extension to $\tau$-leaping and is proposed to improve accuracy by specifically considering how the rate matrix changes according to the noise schedule throughout the reverse process. While these $\tau$-leaping variants offer the advantage of parallelization, the inherent approximation error still necessitates using many small steps to achieve high sampling quality.

The recent work for JYS [17] attempts to accelerate the sampling process of DDMs by focusing on optimizing the time steps of the sampling schedule. It minimizes a Kullback–Leibler divergence upper bound (KLUB) that implicitly captures the overall impact of the compounding decoding error and strategically allocates sampling steps. However, JYS operates by optimizing when to sample, rather than how to sample. At each chosen time, it still relies on the intrinsically biased model and employs standard large-step approximations that suffer from a significant discretization error.

## 1.2   Contributions

To address limitations mentioned above, we move beyond fixed or hand-tuned inference strategies. We introduce a novel learnable sampler distillation (LSD) approach specifically for DDMs. We employ a teacher sampler using small step sizes to approximate a high-quality trajectory. A student sampler is then trained with larger step sizes. Instead of mimicking only the final output of the teacher sampler, which is challenging due to non-differentiability in the discrete pipeline, the student sampler learns to align its intermediate score trajectory with that of the teacher sampler. This alignment is achieved by optimizing the learnable sampler coefficients, which provide the ability to adaptively adjust the sampling dynamics at each step and potentially compensate for the accumulated error given larger step sizes. Furthermore, we propose LSD+ that also learns sampling time schedules. This is done by comparing the effective transition term in the reverse process at intermediate stages and empirically works better compared with uniform sampling schedules used by LSD. We also utilize a relaxed objective during the learning process to alleviate the difficulty of hard alignment between the teacher and student samplers.

Overall, our contribution can be summarized as follows:

- We propose the LSD approach. Inspired by the insight of aligning intermediate score trajectories, LSD trains an efficient student sampler via distillation by optimizing learnable sampler coefficients and incorporating a relaxed training objective for improved feasibility.
- We further introduce LSD+, an extension to LSD that additionally learns non-uniform time schedules. This allows for adaptive allocation of sampling steps, offering a mechanism to potentially better capture varying dynamics and further reduce accumulated errors compared to using uniform time schedules.
- Extensive experiments across text generation, image generation, and synthetic data tasks demonstrate that our proposed approaches achieve significantly higher sampling quality compared to existing baselines at reduced NFEs.

## 2 Preliminaries

### 2.1 Continuous time discrete diffusion models

DDMs model the generative process that can be expressed as a continuous time Markov chain (CTMC) on a finite state space $\mathcal{X} = \{1, \ldots, N\}$ [8, 42]. The forward process describes how data is corrupted. Specifically, the probability of transitioning from state $x$ at time $t$ to state $y$ after a small time interval $\Delta t$ is denoted by $p_{t+\Delta t|t}(y|x)$. This is characterized by [9]:

$$p_{t+\Delta t|t}(y|x) = \begin{cases} Q_t(x, y)\Delta t + o(\Delta t), & y \neq x, \\ 1 + Q_t(x, x)\Delta t + o(\Delta t), & y = x, \end{cases} \tag{1}$$

where $Q_t(x, y)$ is the $(x, y)$ element of the transition rate matrix $Q_t$. The transition rate matrix $Q_t$ is usually formed as $\sigma(t)Q$ [8], where $\sigma(t)$ is a scalar factor, $Q$ is a pre-defined standard matrix with special structures [8]. Let $p_t$ denote the marginal distribution of states at time $t$. In particular, $p_0 = p_{\text{data}}$ is the true distribution of the data. Additionally, for the terminal time $T$, $p_T$ approaches a distribution $\pi$. Depending on $Q$, $\pi$ can mainly be modeled as two distributions, namely a uniform distribution or a distribution that converts samples into masked tokens. For the reverse process that transfers $p_T$ back to $p_0$, the inverse CTMC can be characterized as follows [9]:

$$p_{t-\Delta t|t}(y|x) = \begin{cases} \tilde{Q}_t(x, y)\Delta t + o(\Delta t), & y \neq x, \\ 1 + \tilde{Q}_t(x, x)\Delta t + o(\Delta t), & y = x, \end{cases} \tag{2}$$

where $\tilde{Q}_t$ is the reverse transition rate matrix [41], which can be parameterized by:

$$\tilde{Q}(x, y) = \begin{cases} \frac{p_t(y)}{p_t(x)} Q_t(y, x), & y \neq x, \\ -\sum_{z \neq x} \tilde{Q}_t(x, z), & y = x. \end{cases} \tag{3}$$

The concrete score term $\frac{p_t(y)}{p_t(x)}$ needs to be estimated, as $p_t$ is generally unknown. Therefore, the goal of training a score network $s_\theta : \mathcal{X} \times \mathbb{R} \to \mathbb{R}^{|\mathcal{X}|}$ is to approximate these score values. For instance, SEDD [6] provides an effective method that learns $s_\theta$ such that it satisfies $s_\theta(x, t) \approx \left[\frac{p_t(y)}{p_t(x)}\right]_{y \neq x}$.

## 3 Methods

Achieving efficient inference in DDMs with high sampling quality necessitates accurate approximation of the reverse CTMC using significantly fewer steps than traditional high-fidelity samplers. Numerical samplers like the Euler sampler approximate this process by taking discrete steps guided by the concrete score from the model. However, increasing the step size for faster inference alters the discrete transition dynamics, leading to increased accumulated errors and degraded sampling quality. To enable accurate sampling with large step sizes, we propose learnable sampler distillation (LSD) to make some components of the numerical sampler learnable. Specifically, LSD employs learnable coefficients to dynamically adjust the influence of the concrete score at each time step, allowing the sampler to compensate for large-step discretization errors. Furthermore, while LSD could lead to significant improvement in the generation quality, we further propose LSD+. Instead of learning the coefficients using a fixed uniform schedule, LSD+ further learns a sequence of non-uniform time steps. By training these parameters through distillation from a high-quality teacher sampler, our method learns an optimized discrete-time trajectory. Figure 1 shows the pipeline of our method, and the details of the method are described in the following subsections.

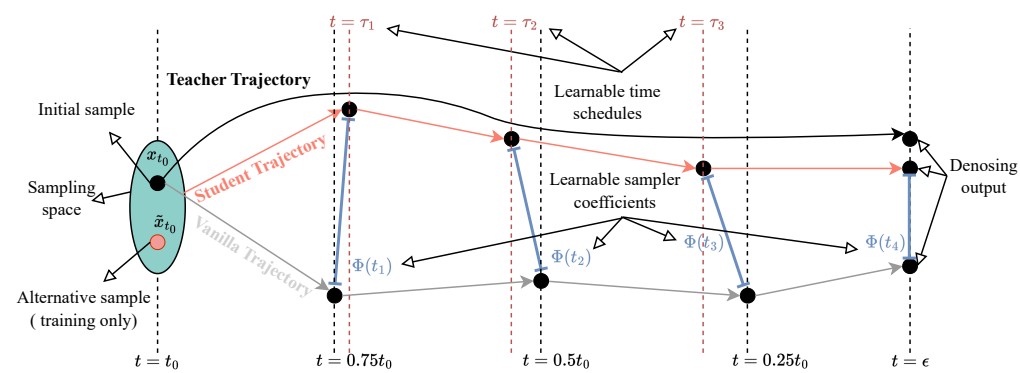

Figure 1: Illustration of our LSD+ approach. During training, the student sampler starts from an alternative initial sample $\tilde{x}_{t_0}$ within the sample space (close to the original $x_{t_0}$). During inference, sampling starts from the initial sample $x_{t_0}$. We can see that the vanilla sampling trajectory often introduces significant discretization errors with large step sizes. In contrast, LSD+ employs learnable coefficients $\Phi(t_k)$ and learnable time schedules $\tau_k$ to adaptively adjust its trajectory. This enables the LSD sampler to more accurately mimic the trajectory of the teacher sampler, effectively compensating for errors inherent in accelerated sampling.

## 3.1 LSD: Coefficients

In this subsection, we illustrate our LSD approach that learns time-dependent coefficients for accelerating DDMs. Given a pre-trained score network $s_\theta(\cdot, \cdot)$, transition rate matrix $Q_t$, and an initial state $x_T$ sampled from $p_T$ at the initial time $T$, the reverse sampling process for DDMs generates a sample by iteratively applying an update rule. For an Euler-type sampler, the transition probability for the $i$-th token from its current state $x_t$ at time $t$ to the next state $x_{t-\Delta t}$ at time $t - \Delta t$ can be parameterized as:

$$p(x_{t-\Delta t}^i | x_t^i) = \delta_{x_t^i}(x_{t-\Delta t}^i) + \Delta t \, Q_t(x_t^i, x_{t-\Delta t}^i) \, s_\theta(x_t, t)_{i, x_{t-\Delta t}^i}. \tag{4}$$

Here, $x_t^i$ denotes the $i$-th token of the current state sequence $x_t$, $\delta_{x_t^i}(x_{t-\Delta t}^i)$ is the Kronecker delta function, $\Delta t$ represents the time step size, $Q_t(x_t^i, x_{t-\Delta t}^i)$ denotes the $(x_t^i, x_{t-\Delta t}^i)$ element of the transition rate matrix $Q_t$, and $s_\theta(x_t, t)_{i, x_{t-\Delta t}^i}$ is the $(i, x_{t-\Delta t}^i)$ element of the concrete score $s_\theta(x_t, t)$.

We apply a fixed teacher sampler that approximates the true reverse process with high fidelity using time schedules $\{t_j^*\}_{j=0}^N$ comprising $N$ steps, with $T = t_0^* > t_1^* > \cdots > t_N^* = \epsilon > 0$.[2] The state generated by the teacher sampler at time $t_j^*$ along its trajectory is denoted as $x_{t_j}^*$. The sampling process of a teacher sampler $\Psi^*$ yields a high-quality final sample $x_\epsilon^* = \Psi^*\left(x_T, \{t_j^*\}_{j=0}^N, s_\theta, \{Q_{t_j^*}\}_{j=0}^N\right)$, which is abbreviated as $x_\epsilon^* = \Psi^*(x_T)$ for simplicity in notation.

We apply a student sampler that operates with a time schedule $\{t_k\}_{k=0}^M$ comprising $M$ steps, where $M \ll N$ and $T = t_0 > t_1 > \cdots > t_M = \epsilon > 0$. $\{t_k\}_{k=0}^M$ is a subsequence of $\{t_j^*\}_{j=0}^N$. Our goal is to learn a set of time-dependent coefficients $\Phi = \{\Phi(t_k)\}_{k=1}^M$ to improve the quality of the output of the student sampler.[3] The state generated by the student sampler at time $t_k$ along its trajectory is denoted as $x_{t_k}$. The sampling process of a student sampler $\Psi$ yields a final sample $x_\epsilon = \Psi\left(x_T, \{t_k\}_{k=0}^M, s_\theta, \{Q_{t_k}\}_{k=0}^M, \{\Phi(t_k)\}_{k=1}^M\right)$, which is abbreviated as $x_\epsilon = \Psi_\Phi(x_T)$ to highlight the dependence on the coefficients $\{\Phi(t_k)\}_{k=1}^M$.

The update rule for the $i$-th token within the student sampler incorporating $\Phi$ becomes:

$$p(x_{t_{k+1}}^i | x_{t_k}^i) = \delta_{x_{t_k}^i}(x_{t_{k+1}}^i) + \Delta t \, Q_{t_k}(x_{t_k}^i, x_{t_{k+1}}^i) \, (\Phi(t_k) s_\theta(x_{t_k}, t_k))_{i, x_{t_{k+1}}^i}. \tag{5}$$

Similar to strategies in some learning methods for continuous ODE solvers [32], a direct objective is to minimize the distance $\mathrm{d}(x_\epsilon, x_\epsilon^*)$, where $\mathrm{d}(\cdot, \cdot)$ is a certain distance metric. However, it is

---

[2]Here, $\epsilon$ is a positive value close to 0 to avoid stability issues as discussed in [4].

[3]Here, $\Phi(t_0)$ is fixed to 1.

generally infeasible for DDMs to directly minimize $\mathrm{d}(x_\epsilon, x_\epsilon^*)$ since the non-differentiable categorical sampling at each step obstructs gradient propagation. Instead, we propose to align intermediate score predictions. At each time step $t_k$, the student sampler computes its score $s_k = s_\theta(x_{t_k}, t_k)$. The teacher sampler evolves its state to the same time step $t_k$ (i.e., for certain $j$ such that $t_j^* = t_k$) using its more accurate sampling process and caches its score $s_k^* = s_\theta(x_{t_k}^*, t_k)$. The states $x_{t_k}$ and $x_{t_k}^*$ differ due to the distinct sampling paths taken to reach $t_k$. Then, our objective is to minimize the discrepancy between $s_k^*$ and $\Phi(t_k)s_k$ for all $k \in \{1, 2, \ldots, M\}$. This can be expressed as:

$$\mathcal{L}_k(\Phi(t_k)) = \mathbb{E}_{x_{t_0} \sim \pi} \left[ \mathrm{d}\left( s_k^*, \Phi(t_k)s_k \right) \right]. \tag{6}$$

This intermediate score trajectory alignment provides a differentiable path for optimizing $\{\Phi(t_k)\}_{k=1}^M$ and ensures the student sampler mimics the trajectory of the teacher sampler across the full denoising path, not just at the final output. We present details of the sampling and training processes for LSD in Algorithms 1 and 2 respectively.

### 3.2 LSD+: Coefficients with learnable time schedules

While LSD improves the sampler by learning the sequence of coefficients $\{\Phi(t_k)\}_{k=1}^M$ under a fixed time schedule, the reverse diffusion dynamics vary significantly across time. We additionally propose LSD+ to also learn non-uniform time schedules. The intuition is that, by learning from a high-fidelity teacher sampler, the student sampler implicitly learns to allocate its limited steps in a manner that best approximates the trajectory of the teacher. Specifically, given time steps for the student sampler $\{t_k\}_{k=0}^M$, the uniform time schedule uses a step size at $\Delta t = \frac{T-\epsilon}{M}$. Our goal is to learn customized step sizes $\{\kappa_k\}_{k=1}^M$, which are initialized as $\Delta t$. The learnable time steps are calculated by:

$$\tau_k = T - \sum_{\ell=1}^k \kappa_\ell. \tag{7}$$

At each learned time step $\tau_k$, the student sampler computes its score $s_\theta(x_{\tau_k}, \tau_k)$. To learn the step size $\kappa_k$, we utilize the so-called effective transition term in the reverse process. Specifically, for the student sampler, this is proportional to $\kappa_k s_\theta(x_{\tau_k}, \tau_k)$, for the teacher sampler, this is proportional to $\frac{T-\epsilon}{N} s_\theta(x_{t_k}^*, t_k)$, where $\frac{T-\epsilon}{N}$ is the step size for the teacher sampler and $s_\theta(x_{t_k}^*, t_k)$ is the cached score of teacher sampler. By calculating the distance of the effective transition terms between the student sampler and teacher sampler, we effectively update $\kappa_k$ considering the unique characteristics of DDMs. This allows the time schedule to adaptively allocate step sizes based on the specific transition structures in DDMs. The updating process can be parameterized as follows:

$$\tilde{\mathcal{L}}_k(\kappa_k) = \mathbb{E}_{x_{t_0} \sim \pi} \left[ \mathrm{d}\left( \kappa_k s_\theta(x_{\tau_k}, \tau_k), \frac{T-\epsilon}{N} s_\theta(x_{t_k}^*, t_k) \right) \right]. \tag{8}$$

We present details of the training and sampling processes for LSD+ in the supplementary material.

---

**Algorithm 1** Sampling process of LSD

---

**Require:** Score network $s_\theta$, time schedule $\{t_k\}_{k=0}^M$ for the student sampler, learned coefficients of the student sampler $\{\Phi(t_k)\}_{k=1}^M$, transition rate matrices $\{Q_{t_k}\}_{k=0}^M$
1: Sample $x_{t_0} \sim \pi$
2: **for** $k = 0$ to $M - 1$ **do**
3:     Sample $x_{t_{k+1}}$ based on $x_{t_k}$ and $\Phi(t_k)$:
4:        $p(x_{t_{k+1}}^i | x_{t_k}^i) = \delta_{x_{t_k}^i}(x_{t_{k+1}}^i) + (t_k - t_{k+1}) Q_{t_k}(x_{t_k}^i, x_{t_{k+1}}^i) \left( \Phi(t_k)s_\theta(x_{t_k}, t_k) \right)_{i, x_{t_{k+1}}^i}$
5:        $x_{t_{k+1}}^i \sim p(x_{t_{k+1}}^i | x_{t_k}^i)$ for all $i$
6: **end for**
7: **return** $x_\epsilon$

---

**Algorithm 2** Training process of LSD

---

**Require:** Score network $s_\theta$, frozen teacher sampler $\Psi^*$ with $N$ steps, learnable student sampler $\Psi_\Phi$ with $M$ steps, learning rate $\eta$, distance metric d, time schedule $\{t_j^*\}_{j=0}^N$ for the teacher sampler, time schedule $\{t_k\}_{k=0}^M$ for the student sampler (a subsequence of $\{t_j^*\}_{j=0}^N$), transition rate matrices $\{Q_{t_j^*}\}_{j=0}^N$

1: Initialize $\Phi(t_k) = 1$ for $k = 1, \ldots, M$
2: **while** not converged **do**
3:     Sample $x_{t_0} \sim \pi$, set $x_{t_0}^* \leftarrow x_{t_0}$
4:     **for** $k = 1$ to $M$ **do**
5:         Calculate the state $x_{t_k}$ generated by the student sampler at time $t_k$ and calculate the score $s_k = s_\theta(x_{t_k}, t_k)$
6:         Calculate the state $x_{t_k}^*$ generated by the teacher sampler at time $t_k$ and calculate the score $s_k^* = s_\theta(x_{t_k}^*, t_k)$
7:     **end for**
8:     **for** $k = 1$ to $M$ **do**
9:         $L_k \leftarrow \mathrm{d}(\Phi(t_k)s_k, s_k^*)$
10:        $\Phi(t_k) \leftarrow \Phi(t_k) - \eta\nabla_{\Phi(t_k)}L_k$
11:     **end for**
12: **end while**
13: **return** $\{\Phi(t_k)\}_{k=1}^M$

---

## 3.3 Relaxed objective

For a student sampler which typically has lower NFEs compared to a teacher sampler, it is non-trivial to force it to accurately match the output of the teacher sampler given the same initial input $x_{t_0}$. Thus, we adopt a relaxed training objective for both LSD and LSD+. We take LSD as an example for further presentation. Instead of strictly requiring the score of the student sampler $s_\theta(x_{t_0}, t_0)$ to match the score of the teacher sampler $s_\theta(x_{t_0}^*, t_0)$, we only require that there exists an alternative input $\tilde{x}_{t_0}$ sufficiently close to the original $x_{t_0}$ (within a small Hamming distance [43] in our discrete token space). Specifically, $\tilde{x}_{t_0}$ satisfies:

$$\mathrm{d_H}(x_{t_0}, \tilde{x}_{t_0}) \leq \zeta, \tag{9}$$

where $\mathrm{d_H}(\cdot, \cdot)$ denotes the Hamming distance between two sequences, $\zeta$ represents positive integer threshold that defines the maximum allowed Hamming distance between $\tilde{x}_{t_0}$ and $x_{t_0}$, where we set it as around 5% of the sequence length.

Therefore, the output of the student sampler at this perturbed score should approximately match the score of the teacher sampler at the original input such that $s_\theta(\tilde{x}_{t_0}, t_0) \approx s_\theta(x_{t_0}^*, t_0)$. Moreover, the relaxed objective function for LSD can be expressed as:

$$\mathcal{L}_{\text{relaxed},k}(\Phi(t_k)) = \mathbb{E}_{x_{t_0}, \tilde{x}_{t_0}} \left[ \mathrm{d}\left( s_\theta(x_{t_k}^*, t_k), \Phi(t_k)s_\theta(\tilde{x}_{t_k}, t_k) \right) \right], \tag{10}$$

where $\tilde{x}_{t_0}$ and $x_{t_0}$ satisfies Eq. (9) and $\tilde{x}_{t_k}$ is sampled starting from $\tilde{x}_{t_0}$. This relaxation makes the optimization task more feasible for the capacity-constrained student sampler by alleviating the rigorous matching requirement. Notably, this input perturbation $\tilde{x}_{t_0}$ is only used during training, at inference time, the student sampler receives the original and unperturbed input $x_{t_0}$. We provide further discussion on the reasonableness of the relaxed objective in the supplementary material.

## 4 Experiments

In this section, we empirically evaluate the performance of our proposed LSD approach and its improved version LSD+. Our goal is to validate their ability to generate high-quality samples at low NFEs. We conduct evaluations across diverse settings, including text generation, image generation, and a synthetic sequence task, comparing against various baselines. We highlight that our LSD+ provides an efficient learning process for the coefficients and time schedules, typically requiring 5 minutes on an NVIDIA RTX4090 GPU, compared to around 10 minutes of training time for JYS under the same environment. And the learned student sampler introduces no additional computational burden during sampling.

## 4.1 Text generation

For the text generation task, we employed three pre-trained DDM backbones for validation, namely SEDD-small [6], SEDD-medium [6], and RADD [9]. These are absorbing DDMs of GPT-2 level for text generation, trained on the OpenWebText dataset [44]. For the uniform DDMs, please refer to the supplementary material. We compare LSD and LSD+ against standard Euler and Tweedie samplers [6] and the JYS method [17]. For the RADD baseline, we also compare with higher-order samplers, the $\theta$-RK-2 and $\theta$-trapezoidal [45].[4] We evaluate the generative perplexity of unconditionally generated text using a GPT2-large model. We generated 1024 samples, each containing 1024 tokens. The results are presented in Tables 1, 2, and 3. LSD (LSD+)-Euler (Tweedie) denotes that we implement LSD (LSD+) based on the Euler (Tweedie) sampler [6]. The empirical results show that our methods significantly outperform the baseline methods across all three backbones and all tested NFEs. Moreover, we find that LSD+ generally outperforms LSD, which indicates that the learned non-uniform time schedules help to further reduce accumulated errors. Therefore, we only present the results for LSD+ for the experiments in Sections 4.2 and 4.3.

Table 1: Comparison of generative perplexity ($\leq$) on the SEDD-small backbone. Best performances are bolded.

| NFEs
Sampler | 8 | 16 | 32 | 64 |
|---|---|---|---|---|
| Euler | 423.109 | 215.472 | 72.820 | 56.218 |
| Tweedie | 404.881 | 177.539 | 64.347 | 50.151 |
| JYS-Euler | 308.123 | 125.283 | 55.842 | 32.943 |
| JYS-Tweedie | 307.382 | 127.232 | 56.382 | 31.192 |
| LSD-Euler | 145.490 | 88.564 | **31.235** | 21.956 |
| LSD-Tweedie | 168.846 | 86.282 | 35.786 | 21.981 |
| LSD+-Euler | **128.413** | **51.769** | 36.800 | 20.728 |
| LSD+-Tweedie | 137.862 | 60.970 | 38.157 | **20.473** |

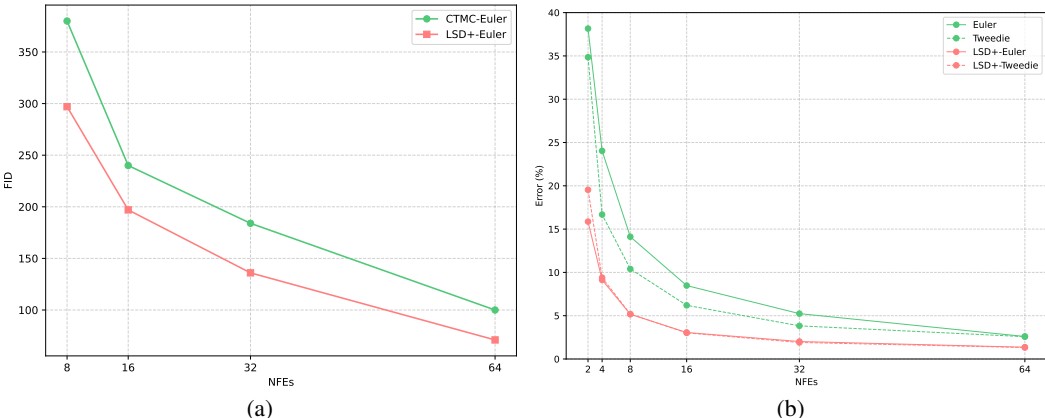

Figure 2: Validation on (a) the image generation task and (b) the synthetic countdown task. Our LSD+ method shows superior performance.

## 4.2 Image generation

We also validate our LSD+ approach on the image generation task for the CIFAR-10 dataset [46]. We utilize CTMC [8] as the baseline, which employs a Gaussian transition matrix and denoising parameterization. Each data sample is a flattened image with a size of $3 \times 32 \times 32$, composed of

---

[4]Since the source code of [45] is inaccessible, we could only compare our method with these high-order samplers on the RADD backbone. The results are reported from the paper. "/" in Table 3 denotes that this paper does not report the results when NFEs is 8.

Table 2: Comparison of generative perplexity ($\leq$) on the SEDD-medium backbone. Best performances are bolded.

| NFEs Sampler | 8 | 16 | 32 | 64 |
|---|---|---|---|---|
| Euler | 399.315 | 184.603 | 77.925 | 44.370 |
| Tweedie | 394.470 | 178.485 | 67.114 | 40.487 |
| JYS-Euler | 299.394 | 115.853 | 43.958 | 25.545 |
| JYS-Tweedie | 300.492 | 118.218 | 48.430 | 28.539 |
| LSD-Euler | 125.769 | 54.865 | 28.001 | 19.886 |
| LSD-Tweedie | 98.209 | 51.223 | 26.668 | 20.794 |
| LSD+-Euler | 121.583 | **46.145** | **25.144** | **15.929** |
| LSD+-Tweedie | **90.033** | 50.765 | 26.799 | 16.239 |

Table 3: Comparison of generative perplexity ($\leq$) on the RADD backbone. Best performances are bolded.

| NFEs Sampler | 8 | 16 | 32 | 64 |
|---|---|---|---|---|
| Euler | 670.977 | 282.115 | 152.403 | 113.913 |
| Tweedie | 648.736 | 285.471 | 155.472 | 98.879 |
| $\theta$-RK2 | / | 127.363 | 109.351 | 66.549 |
| $\theta$-Trapezoidal | / | 123.585 | 89.912 | 66.549 |
| LSD-Euler | 121.420 | 60.219 | 46.268 | 32.817 |
| LSD-Tweedie | 122.167 | 67.176 | 40.274 | 28.644 |
| LSD+-Euler | **89.830** | **36.106** | **33.234** | 29.115 |
| LSD+-Tweedie | 90.364 | 40.263 | 36.129 | **24.312** |

tokens with values ranging from 0 to 255. We evaluate the FID score using 50k samples with the NFEs selected from $\{8, 16, 32, 64\}$. Figure 2(a) shows the results and we can observe that our method provides better FID scores compared to the baseline method.

### 4.3 Synthetic countdown task

We follow [47] to evaluate our LSD+ approach on a synthetic sequence task with strong dependencies. The dataset features 256-token sequences (with values in 0-31) where non-zero tokens must strictly decrease by one. We trained an absorbing SEDD [6] model and measured performance by the error rate, which is the proportion of generated samples violating this countdown rule. As shown in Figure 2(b), our method achieves lower error rates across various NFEs compared to baselines.

## 5 Ablation Study

The learnable coefficients form the core of our LSD approach and have demonstrated significant performance improvements. Also, we observe that LSD+ generally outperforms LSD as seen in Tables 1, 2, and 3, which indicates the benefit of the learned non-uniform time schedules. Therefore, our ablation study aims to assess the contributions of the relaxed objective during training.

**Benefit of the relaxed objective** We proposed a relaxed objective, allowing the student sampler to match the trajectory of the teacher sampler originating from $x_{t_0}$ by using a perturbed starting point $\tilde{x}_{t_0}$ that is close to $x_{t_0}$ during the training process. Table 4 compares the performance of LSD+ trained with and without this relaxation. The results clearly indicate that employing the relaxed objective generally yields better performance than training with the strict objective. This validates the benefit of the relaxation, confirming that it makes the trajectory alignment task more feasible and leads to better convergence.

**Impact of Hamming distance threshold** To investigate the robustness of the algorithm to the Hamming distance threshold, we conduct the ablation on the SEDD-small backbone using the Euler

sampler with 32 inference steps. We train our LSD+ method using several different values for the Hamming distance threshold, specifically 0%, 1%, 5%, 10%, 20% of the sequence length, while keeping all other hyperparameters unchanged. The performance, measured by Perplexity, is reported below in Table 5.

Table 4: Ablation study on the RADD backbone validating the importance of the Relaxed Objective (RO). "w/o RO" indicates parameters learned without relaxed objective, while "w/ RO" denotes parameters learned using our proposed relaxed objective. Both settings are evaluated using either Euler or Tweedie as the base sampler for our LSD+ method. Best performances are bolded.

| NFEs Sampler | 8 | 16 | 32 | 64 |
|---|---|---|---|---|
| LSD+ w/o RO-Euler | 95.943 | 38.192 | 34.983 | 31.392 |
| LSD+ w/ RO-Euler | **89.830** | **36.106** | **33.234** | **29.115** |

Table 5: Ablation study on the Hamming distance threshold for the relaxed objective.

| Threshold(%) | 0 | 1 | 5 (Our choice) | 10 | 20 |
|---|---|---|---|---|---|
| Perplexity($\downarrow$) | 35.98 | 32.15 | **31.24** | 39.97 | 51.52 |

# 6 Conclusion

This paper aims to address the challenge of inefficient sampling in DDMs, a major obstacle to their practical deployment. While reducing the NFEs accelerates inference, previous accelerating methods suffer from accumulated compounding decoding error and discretization errors, significantly degrading sampling quality. We introduce LSD, a novel approach that leverages distillation from a high-fidelity teacher sampler. Instead of merely matching final outputs, LSD trains a student sampler with a few steps to align its entire intermediate score trajectory with that of the teacher sampler. This is achieved by optimizing learnable, time-dependent coefficients. And we additionally propose LSD+ that also learns non-uniform sampling schedules and this allows the sampler to adaptively compensate for errors induced by larger step sizes. Extensive experiments demonstrate that our methods significantly outperform the baseline samplers across diverse tasks, achieving high sampling fidelity at low NFEs.

A promising direction for future research is to provide theoretical guarantees regarding the distributional discrepancy between the outputs of teacher and student samplers, potentially building on existing theoretical findings related to discrete diffusion models [48, 49, 50, 51, 52, 45, 53, 54].

**Acknowledgment.** We sincerely thank the five anonymous reviewers and the area chair for their meticulous reading of our work and their constructive comments, which have substantially enhanced the quality and rigor of our study.

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

# Supplementary Material

## Learnable Sampler Distillation for Discrete Diffusion Models (NeurIPS 2025)

Feiyang Fu, Tongxian Guo, Zhaoqiang Liu

## A  Limitations and Broader Impact

While our proposed learnable sampler distillation (LSD) approach significantly enhances the sampling quality of discrete diffusion models (DDMs) at low NFEs, there exists a limitation that warrants discussion. Specifically, the performance ceiling of the student sampler is inherently tied to the quality of the teacher sampler. Consequently, while our student sampler can efficiently achieve the performance of the teacher sampler with very few NFEs, surpassing it is challenging. However, this dependence on the teacher is an intrinsic characteristic of knowledge distillation paradigms and LSD is compatible with future advancements, since more sophisticated samplers can serve as improved teacher samplers.

Our approaches can be integrated with various DDM samplers, offering a general path to enhance their efficiency, and hold significant promise for accelerating scientific discovery, such as by facilitating the design of candidate DNA and protein sequences. However, the increased power of generative AI also necessitates a commitment to responsible development. This includes proactive efforts to mitigate societal risks, notably the potential for generating misinformation or amplifying existing biases, and underscores the importance of ethical guidelines and detection research.

## B  Discussion

### B.1  Intuitive explanation

Given that several theoretical works on DDMs have been conducted [49, 55, 56, 48], we provide an intuitive explanation for the effectiveness of our method, which may help lay the groundwork for subsequent theoretical framing. Specifically, the final discrepancy between the outputs of the student and teacher samplers stems from the accumulation of small local errors made at each step. Each of these local errors, in turn, is tied to the specific score predicted by the model at that step. Our training objective directly enforces alignment between the score predictions of the student sampler and those of the teacher sampler at every step. By implicitly correcting these small local errors throughout the process, our approach guides the student sampler to produce final outputs that closely match the high-quality results of the teacher sampler.

### B.2  Relaxed objective

Our use of a relaxed training objective enhances training feasibility compared to strict alignment. However, the theoretical guarantees that the resulting student distribution closely matches the teacher distribution might be less rigorous than what could potentially be argued for in continuous spaces. To the best of our knowledge, establishing such rigorous theoretical bounds in discrete spaces faces several challenges not present in their continuous counterparts. Specifically, (1) In continuous diffusion, initial state perturbations can often be controlled and analyzed via differentiable operations (e.g., $L_2$-constrained gradient steps) [32]. In contrast, discrete initial states (e.g., token sequences) and their perturbations that are measured by Hamming distance lack this differentiability. This precludes similar continuous optimization and analysis pathways. (2) The analytical tools used for relaxed objective in continuous diffusion ODE/SDE, such as perturbation analysis for smooth dynamical systems [31], do not directly translate to the discrete dynamics of CTMCs and their approximations. Therefore, our proposed relaxed objective serves as an empirically effective solution for training feasibility in LSD.

## C Details for discrete diffusion models

In the main text, we illustrate the mechanism by which the forward process of CTMC introduces corruption into the data. Within Eq. (1), the term $Q_t(x, y)$ is delineated as the transition rate from state $x$ to state $y$ at time $t$. Consistent with this definition, $Q_t(x, y)$ can be articulated as follows:

$$Q_t(x,y) = \begin{cases} \lim_{\Delta t \to 0} \frac{p_{t+\Delta t|t}(y|x)}{\Delta t}, & y \neq x, \\ \lim_{\Delta t \to 0} \frac{p_{t+\Delta t|t}(x|x)-1}{\Delta t}, & y = x. \end{cases} \tag{11}$$

For instances where $t > s$, we define $P_{t|s}(x, y) := p_{t|s}(y|x)$. Drawing upon Kolmogorov's forward equation [8] and the definition of $Q_t$, we derive

$$\frac{d}{dt}P_{t|s} = P_{t|s}Q_t. \tag{12}$$

The analytical solution to Eq. (12) is given by $P_{t|s} = \exp\left((\bar{\sigma}(t) - \bar{\sigma}(s))Q\right)$, with $\bar{\sigma}(t)$ denoting the integral $\int_0^t \sigma(s)ds$ and $\exp$ denoting the matrix exponential function. This solution facilitates the direct sampling of $x_t$ from $x_s$ in a single step for all $t > s$ scenarios.

The transition rate matrix $Q_t$ is formulated as $\sigma(t)Q$, where Q represents a pre-specified standard matrix. In the context of defining matrix $Q$, two principal alternatives are presented: A uniform distribution or a MASK absorbing state. When the base transition matrix $Q$ is selected to be a uniform matrix, it simulates a fully connected graph structure. Within this framework, each state is interconnected with all other states, implying that transitions from any given state to any other state are possible. The definition of this transition matrix ensures extensive exploratory capabilities of the state space, permitting the model to account for all potential state transitions during simulation. Specifically, the construction of $Q^{\text{uniform}}$ is as follows:

$$Q^{\text{uniform}} = \begin{bmatrix} 1-N & 1 & \cdots & 1 & 1 \\ 1 & 1-N & \cdots & 1 & 1 \\ \vdots & \vdots & \ddots & \vdots & \vdots \\ 1 & 1 & \cdots & 1-N & 1 \\ 1 & 1 & \cdots & 1 & 1-N \end{bmatrix}. \tag{13}$$

An alternative approach in the construction of the transition matrix $Q$ includes the formulation of a matrix that encompasses absorbing states. An absorbing state refers to a state, where the system will no longer undergo state transitions once it enters this state, and such a design facilitates the rapid convergence of the model to a stable state. Recent works [6, 9] have also demonstrated that the adoption of an absorbing matrix is associated with better performance and serves to accelerate the sampling process. The construction of $Q^{\text{absorb}}$ is defined as:

$$Q^{\text{absorb}} = \begin{bmatrix} -1 & 0 & \cdots & 0 & 1 \\ 0 & -1 & \cdots & 0 & 1 \\ \vdots & \vdots & \ddots & \vdots & \vdots \\ 0 & 0 & \cdots & -1 & 1 \\ 0 & 0 & \cdots & 0 & 0 \end{bmatrix}. \tag{14}$$

To effectively simulate the reverse process, the common practice is to train a neural network $s_\theta(x, t)$ to approximate the required concrete score $\frac{p_t(y)}{p_t(x)}$. In this process, to optimize the training of network $s_\theta(x, t)$, SEDD [6] introduces an effective loss function:

$$\int_0^T \mathbb{E}_{x \sim p_{t|0}(x|x_0)} \sum_{y \neq x} Q_t(y, x) \left( s_\theta(x, t)_y - \frac{p_{t|0}(y \mid x_0)}{p_{t|0}(x \mid x_0)} \log s_\theta(x, t)_y + C \right) dt, \tag{15}$$

where $C$ is a constant with $C = K\left(\frac{p_{t|0}(y|x_0)}{p_{t|0}(x|x_0)}\right)$ and $K(a) := a \log a - a$, $s_\theta(x, t)_y$ represents an estimate by the neural network of the probability of transitioning from state $x$ to state $y$ at time $t$.

# D Additional method details

## D.1 LSD using Tweedie $\tau$-leaping

In the main text, we detailed the application of our LSD appoach to an Euler-type discrete sampler. Notably, LSD can also be readily integrated with other sampling methods, such as Tweedie $\tau$-leaping.

The Tweedie $\tau$-leaping update rule for DDMs leverages Tweedie's formula to relate the conditional expectation of a cleaner state to the score function. Given time schedules $\{t_k\}_{k=0}^M$ for the student sampler. The transition probability $p(x_{t_{k+1}}^i|x_{t_k}^i)$ for the $i$-th token from state $x_{t_k}$ at time $t_k$ to state $x_{t_{k+1}}$ at the subsequent time step $t_{k+1}$ is constructed as follows:

$$
p(x_{t_{k+1}}^i|x_{t_k}^i) = \left(\exp\left((\bar{\sigma}(t_{k+1}) - \bar{\sigma}(t_k))Q\right) s_\theta(x_t, t)_i\right)_{x_{t_{k+1}}^i}
$$
$$
\times \left(\exp\left((\bar{\sigma}(t_k) - \bar{\sigma}(t_{k+1}))Q\right)\right)_{x_t^i, x_{t_{k+1}}^i}, \tag{16}
$$

where $Q$ is the predefined standard matrix with special structure as mentioned in Section C, $\bar{\sigma}(t)$ be the cumulative noise schedule, which is a non-decreasing function of $t$, $s_\theta(x_{t_k}, t_k)_i$ is the $i$-th element of the score $s_\theta(x_{t_k}, t_k)$. The subscript $x_{t_{k+1}}^i$ on the first main term denotes the $x_{t_{k+1}}^i$-th element of the resulting vector. Similarly, the subscript $(x_{t_k}^i, x_{t_{k+1}}^i)$ on the second main term denotes selecting the corresponding element from the second main term.

To incorporate our LSD approach, we introduce the learnable time-dependent coefficient $\{\Phi(t_k)\}_{k=1}^M$ to modulate the score network $s_\theta$ within the Tweedie update. The LSD-modified Tweedie $\tau$-leaping update rule is as follows:

$$
p(x_{t_{k+1}}^i|x_{t_k}^i) = \left(\exp\left((\bar{\sigma}(t_{k+1}) - \bar{\sigma}(t_k))Q\right)(\Phi(t_k)s_\theta(x_{t_k}, t_k))_i\right)_{x_{t_{k+1}}^i}
$$
$$
\times \left(\exp\left((\bar{\sigma}(t_k) - \bar{\sigma}(t_{k+1})Q)\right)\right)_{x_t^i, x_{t_{k+1}}^i}. \tag{17}
$$

This formulation allows the student sampler using Tweedie $\tau$-leaping to learn an adaptive scaling $\{\Phi(t_k)\}_{k=1}^M$ of the score guidance.

## D.2 Pseudocode for LSD+

In this section, we present the sampling and training processes of LSD+ in Algorithms 3 and 4. Empirically, we do not learn the coefficients and time schedules in the same training epoch, which empirically leads to training instability issues. Instead, we learn them separately in different training epochs. For brevity, we only list the algorithm that learns the time schedules, rather than the whole learning process.

---

**Algorithm 3** Sampling process of LSD+

---

**Require:** Score network $s_\theta$, learned time schedule $\{\tau_k\}_{k=0}^M$ for the student sampler with $M$ steps, learned coefficients of the student sampler $\{\Phi(\tau_k)\}_{k=1}^M$, transition rate matrices $\{Q_{\tau_k}\}_{k=0}^M$
1: Sample $x_{t_0} \sim \pi$
2: Sample $x_{\tau_{k+1}}$ based on $x_{\tau_k}$ and $\Phi(\tau_k)$ :
3: $\qquad p(x_{\tau_{k+1}}^i|x_{\tau_k}^i) = \delta_{x_{\tau_k}^i}(x_{\tau_{k+1}}^i) + (\tau_k - \tau_{k+1}) Q_{\tau_k}(x_{\tau_k}^i, x_{\tau_{k+1}}^i) \Phi(\tau_k)(s_\theta(x_{\tau_k}, \tau_k))_{i, x_{\tau_{k+1}}^i}$
4: $\qquad x_{\tau_{k+1}}^i \sim p(x_{\tau_{i+1}}^i|x_{\tau_k}^i)$ for all $i$
5: **return** $x_\epsilon$

---

**Algorithm 4** Training process of LSD+ that learns the time schedules

---

**Require:** Score network $s_\theta$, frozen teacher sampler $\Psi^*$ with $N$ steps, learnable student sampler $\Psi_\Phi$ with $M$ steps, learning rate $\eta$, distance metric d, time schedule $\{t_j^*\}_{j=0}^N$ for the teacher sampler, original time schedule $\{t_k\}_{k=0}^M$ for the student sampler (a subsequence of $\{t_j^*\}_{j=0}^N$), transition rate matrices $\{Q_{t_j^*}\}_{j=0}^N$

1: Initialize step sizes $\kappa_k = \frac{t_0 - t_M}{M}$ for $k = 1, 2, \ldots, M$
2: Initialize learnable time schedule $\tau_k = t_0 - \sum_{\ell=1}^k \kappa_\ell$ for $k = 1, 2, \ldots, M$
3: **while** not converged **do**
4:     Sample $x_{t_0} \sim \pi$, set $x_{t_0}^* \leftarrow x_{t_0}$
5:     **for** $k = 1$ to $M$ **do**
6:         Calculate the state $x_{\tau_k}$ generated by the student sampler at time $\tau_k$ and calculate $s_k = s_\theta(x_{\tau_k}, \tau_k)$
7:         Calculate the state $x_{t_k}^*$ generated by the teacher sampler at time $t_k$ and calculate the score $s_k^* = s_\theta(x_{t_k}^*, t_k)$
8:     **end for**
9:     **for** $k = 1$ to $M$ **do**
10:         $\tilde{L}_k \leftarrow \mathrm{d}(\kappa_k s_k, \frac{t_0^* - t_N^*}{N} s_k^*)$
11:         $\kappa_k \leftarrow \kappa_k - \eta \nabla_{\kappa_k} \tilde{L}_k$
12:         $\tau_k = t_0 - \sum_{l=1}^k \kappa_l$
13:     **end for**
14: **end while**
15: **return** $\{\kappa_k\}_{k=1}^M$

---

# E   Additional empirical details

## E.1   Training settings

In this subsection, we provide training settings and implementation details for reproducing our empirical results. Specifically, we set the terminate time $\epsilon$ as $0.0001$, the total sampling steps $N$ of the teacher sampler as $1024$, and the distance metric d as the KL divergence. We set the number of training samples as $64$, the training epoch as $20$, and the learning rate $\eta$ as $0.001$. All experiments are conducted on an NVIDIA RTX4090 GPU.

## E.2   More results on large-scale image datasets

We conduct experiments on ImageNet (256x256) using the MaskGIT [57] architecture as the backbone, incorporating the recently proposed advanced Halton sampler [58] as the sampling method. The results are reported in Table 6, which demonstrates that our LSD+ method yields improved generation performance as measured by the FID metric. We also study the effect of softmax temperature in MaskGIT on ImageNet (256x256) for the image generation task, with the results listed in Table 7.

Table 6: Comparison on ImageNet 256x256 (in terms of FID↓)

| Sampler \ NFEs | 4 | 8 | 16 | 32 |
|---|---|---|---|---|
| Halton | 14.16 | 10.15 | 8.89 | 6.92 |
| LSD+-Halton | 12.78 | 8.66 | 7.17 | 6.32 |

Table 7: Comparisons on ImageNet 256x256 (in terms of FID↓ when NFE=4).

| Sampler \ Temperature | $\tau = 0.6$ | $\tau = 0.8$ | $\tau = 1.0$ |
|---|---|---|---|
| Halton | 54.05 | 26.45 | 14.16 |
| LSD+-Halton | 48.29 | 24.51 | 12.78 |

### E.3 More results on text generation.

We provide more results on the task of text generation.

We first conduct experiments on predictor-corrector solvers in $\tau$LDR-10 [8] sampler on the SEDD-small backbone, with the results presented in Table 8.

To further break through the limitation of only relying on perplexity to assess generation quality and provide a more thorough evaluation, we conduct experiments on the SDTT-KLD [59] backbone, which utilizes Ancestral as the sampler. We report MAUVE, Perplexity and Entropy scores in Table 9, and these results demonstrate that our LSD+ method usually yields improved generation performance.

Meanwhile, we also integrate LSD+ with ReMasking (ReMDM) [60] on the MDLM [61] backbone, with relevant findings reported in Table 10. The results show that LSD+ can effectively learn to work in conjunction with this method, further improving performance.

To ensure the generalization of our main results, we further re-evaluate our findings on the SEDD and RADD backbones, using Llama-3-8B [62] to assess perplexity. The re-evaluation results are reported in Tables 11 and 12. Moreover, we also integrate our LSD+ with multi-token unmasking heuristics in FastDLLMs [63] and report the corresponding accuracy and throughput metrics in Table 13. These metrics indicate that our methods can improve both sampling quality and speed.

To demonstrate that our method is not limited to smaller models, we conduct experiments on applying LSD+ to larger-scale models including DiffuLLaMA and DiffuGPT [64]. Specifically, we perform sampler distillation on pre-trained DiffuGPT-S and DiffuLLaMA checkpoints, and the results in terms of Perplexity are presented in Tables 14 and 15.

Finally, we conduct experiments on the sampler of DNDM [12], which uses the FairSeq [65] as the backbone, the corresponding perplexity results are reported in Table 16.

Table 8: Comparisons on the SEDD-small backbone.

| Sampler | Perplexity($\downarrow$) | | | | Entropy($\uparrow$) | | | |
|---|---|---|---|---|---|---|---|---|
| NFEs | 16 | 32 | 64 | 128 | 16 | 32 | 64 | 128 |
| $\tau$LDR-10 | 443.17 | 318.44 | 277.16 | 199.51 | 5.63 | 5.69 | 5.57 | 5.24 |
| LSD+$-\tau$LDR-10 | 205.42 | 143.58 | 114.93 | 90.43 | 5.58 | 5.49 | 5.59 | 5.44 |

Table 9: Comparisons on the SDTT-KLD backbone.

| Sampler | MAUVE($\uparrow$) | | | Perplexity($\downarrow$) | | | Entropy($\uparrow$) | | |
|---|---|---|---|---|---|---|---|---|---|
| NFEs | 8 | 16 | 32 | 8 | 16 | 32 | 8 | 16 | 32 |
| Ancestral | 0.884 | 0.912 | 0.943 | 110.391 | 56.652 | 42.128 | 5.331 | 5.285 | 5.222 |
| LSD+-Ancestral | 0.905 | 0.928 | 0.951 | 68.130 | 36.577 | 31.597 | 5.298 | 5.239 | 5.226 |

Table 10: Comparisons on the MDLM backbone.

| Sampler | Perplexity($\downarrow$) | | | | Entropy($\uparrow$) | | | |
|---|---|---|---|---|---|---|---|---|
| NFEs | 16 | 32 | 64 | 128 | 16 | 32 | 64 | 128 |
| ReMDM | 434.08 | 174.72 | 85.15 | 62.33 | 5.73 | 5.66 | 5.48 | 5.55 |
| LSD+-ReMDM | 201.52 | 102.02 | 62.97 | 49.33 | 5.41 | 5.42 | 5.52 | 5.33 |

Table 11: Comparisons on SEDD-small backbone (in terms of Perplexity$\downarrow$), judged by LLaMA-3-8B.

| NFEs / Sampler | 8 | 16 | 32 | 64 |
|---|---|---|---|---|
| Euler | 116.93 | 67.43 | 49.81 | 46.88 |
| LSD+-Euler | 74.65 | 33.25 | 30.70 | 21.64 |

Table 12: Comparisons on RADD backbone (in terms of Perplexity↓), judged by LLaMA-3-8B.

| NFEs
Sampler | 8 | 16 | 32 | 64 |
|---|---|---|---|---|
| Euler | 337.04 | 216.01 | 119.25 | 95.06 |
| LSD+-Euler | 130.00 | 60.90 | 42.53 | 35.65 |

Table 13: Integration with Fast-dLLM on LLaDA. We report accuracy (throughput, token/s).

| Samplers
Benchmark | LLaDA | +Cache | +Parallel | Fast-dLLM | LSD+-Fast-dLLM |
|---|---|---|---|---|---|
| GSM8K(5-shot) | 79.3(6.7) | 79.5(21.2) | 79.2(16.5) | 78.5(54.4) | 79.0(62.5) |
| MATH(4-shot) | 33.5(9.1) | 33.3(23.7) | 33.4(24.8) | 33.2(51.7) | 33.4(58.1) |

Table 14: Comparisons on the DiffuGPT-S backbone (in terms of Perplexity↓).

| NFEs
Method | 16 | 32 | 64 | 128 |
|---|---|---|---|---|
| DiffuGPT-S | 117.32 | 75.19 | 58.34 | 37.16 |
| LSD+-DiffuGPT-S | 53.95 | 41.37 | 32.10 | 22.25 |

Table 15: Comparisons on the DiffuLLaMA backbone (in terms of Perplexity↓).

| NFEs
Method | 16 | 32 | 64 | 128 |
|---|---|---|---|---|
| DiffuLLaMA | 100.04 | 69.11 | 42.17 | 30.55 |
| LSD+-LLaMA | 49.83 | 34.32 | 29.18 | 24.72 |

Table 16: Comparisons on the FairSeq backbone (in terms of Perplexity↓).

| NFEs
Method | 8 | 16 | 32 | 64 |
|---|---|---|---|---|
| DNDM | 919.23 | 774.92 | 748.41 | 622.14 |
| LSD+-DNDM | 601.22 | 554.19 | 477.10 | 403.13 |

## E.4 More results on uniform discrete diffusion models

In the main text, we provide comparisons with existing samplers on the absorbing DDMs. In this subsection, we provide additional empirical results on uniform DDMs in Tables 17, 18 and 19. We oberseve that our methods also outperforms existing sampling methods, validating the superiority and robustness of our methods.

Table 17: Comparison of generative perplexity ($\leq$) on the uniform SEDD-small backbone. Best performances are bolded.

| NFEs
Sampler | 8 | 16 | 32 | 64 |
|---|---|---|---|---|
| Euler | 467.832 | 224.954 | 76.364 | 54.293 |
| Tweedie | 433.590 | 215.233 | 70.361 | 52.364 |
| JYS-Euler | 310.329 | 130.034 | 60.574 | 33.951 |
| JYS-Tweedie | 308.732 | 129.843 | 55.293 | 30.675 |
| LSD-Euler | 160.811 | 103.346 | 38.059 | 23.365 |
| LSD-Tweedie | 177.728 | 101.213 | **37.682** | 25.675 |
| LSD+-Euler | 157.535 | **46.448** | 39.850 | 23.938 |
| LSD+-Tweedie | **128.910** | 47.080 | 38.058 | **21.724** |

Table 18: Comparison of generative perplexity (≤) on the uniform SEDD-medium backbone. Best performances are bolded.

| NFEs / Sampler | 8 | 16 | 32 | 64 |
|---|---|---|---|---|
| Euler | 403.654 | 190.784 | 80.094 | 47.885 |
| Tweedie | 387.743 | 182.312 | 65.853 | 44.754 |
| JYS-Euler | 311.427 | 121.954 | 49.912 | 32.934 |
| JYS-Tweedie | 306.089 | 116.233 | 45.287 | 28.192 |
| LSD-Euler | 132.209 | 60.823 | 28.283 | 22.956 |
| LSD-Tweedie | 118.138 | 53.591 | 27.162 | 21.308 |
| LSD+-Euler | 111.870 | 49.427 | **25.732** | 19.623 |
| LSD+-Tweedie | **93.196** | **47.932** | 27.381 | **18.119** |

Table 19: Comparison of generative perplexity (≤) on the uniform RADD backbone. Best performances are bolded.

| NFEs / Sampler | 8 | 16 | 32 | 64 |
|---|---|---|---|---|
| Euler | 657.732 | 280.743 | 157.825 | 115.743 |
| Tweedie | 652.381 | 277.195 | 160.045 | 108.730 |
| LSD-Euler | 126.394 | 65.923 | 51.197 | 34.764 |
| LSD-Tweedie | 127.098 | 71.034 | 44.936 | 29.834 |
| LSD+-Euler | **94.554** | **41.986** | **32.832** | 27.283 |
| LSD+-Tweedie | 95.029 | 45.823 | 37.900 | **26.883** |

# F  PCA analysis of coefficients

To analyze the the training evolution of learned sampler coefficients $\{\Phi(t_k)\}_{k=1}^{M}$, we performed Principal Component Analysis (PCA) [66, 67] on $\mathbb{R}^M$-dimensional coefficient vectors collected at each epoch. Figure 3 visualizes projections onto the top two principal components. The trajectories of learned coefficients show substantial divergence from the vanilla baseline point, empirically demonstrating that optimization yields configurations distinct from fixed scaling. This dynamic learning process, while sensitive to initialization across runs, explores configurations enabling high-fidelity sampling at low NFEs, and the difference between different training runs is relatively small.

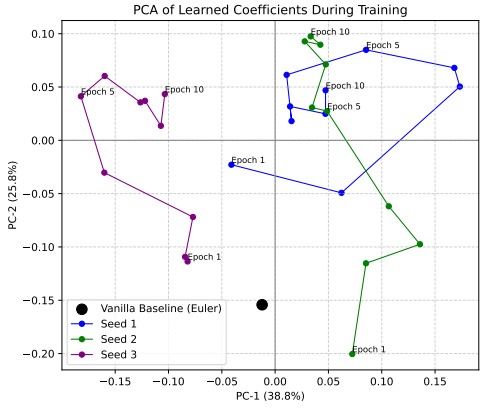

Figure 3: PCA analysis of learned coefficients at each epoch of training.

# G  Text generation results

In this section, we provide text generation results from LSD+. All results are generated using the Euler sampler on the SEDD-medium backbone.

swagqell on Twitter and for reading! Heath and Hello, that's this information in Continue reading. This's a good idea, I've going to have some ask you who. On I asked you to come in a powerful is, with for then, here were about some of the reasons for that.   By any time in Baltimore, in Camden, or Washington, I'll sure you if some sources would   and, well, I.d be in I was in Soltas, of Philadelphia (Washington and elsewhere) about the game that's process and his way on the football team. For instance, cornerback Scott Snyder on Washington called "", the the 49ers' 2012 coach (if "" of those season said more byand (that).   I'm curious if you would ""—to have a pragmatic thing that actually assures a "" the right of the .election.article" level activists and that it's not, to to the system of American columns, to this qualified as follows in a   If I've the. Before the election for the days where the didn't ask, and the Republican candidates to lead at all—without clear direction and without first getting a record in light of the constituents' from which I've taken my time to think about such thing and I'd not be too unapologetic about spending a few days   (Don't know, the rise of the headed system, I's in curious whether he wasn't in. Here thought was in response case. This is the same is of one's attempt to   , so funny enough, long as it remains without a". The 55 lot of the plans, so as more of an in-house plan to than   irmative", a plan to interest increase and saying, that should function as a thought. As for whether Obama won.   , he's currently third in the, in Texas, nominee. I wrote about the new bill in an article-ed. Even if the "If" you believe in him" is. you be the the, which was in favor of repealing it.   In terms of the game being can self-destruct of Romney's spokesperson, Mitt Romney. I was you who did it. case, it was passed last which was the new health insurance law. The law was Obamacare, the American Health Care Act. I think was about to pre the Obamacar

Figure 4: The generated text for NFE=8.

and that the federal programs have been pay out for additional much. It was agreed at the summit that there will be increases in the mean time and that they will need to look back and make full pay."We have got to stop before you get started," Boehner said. "And not a few of us are going to have a budget in the some, because what we have't started looking at that's just beginning. We need to pay for all those programs -- our normal budget plus we will make work toward the end of the year."In order to be trusted in conjunction with the austerity of the entitlement will begin almost immediately. We are the nation -- President Obama will have said it and the Republicans, for that matter, won't have said it -- to have to face.Obama has done everything in his power not to acknowledge this possibility. There maintains a veneer of honestness to his administration, but else he displayed is almost entirely to blame on the cliff.On a united loan from the House Party, Obama has paid—Both parties in ownership the national debt.Specifically: the companies that profit from, and through whose contracts, the money government owes, and it is owed more. It has given a flying strip limit"s. It also has debt small companies, the United States on this earth.In America, the government has had enormous responsibility to America. I can tell Washington DC really has done the least for America over the world.Perhaps it Washington, it will be able to assist and alter the government whether or not they have created something to the taxpayer the financial representation of this debt feels symbolically self sufficient. But their future goes far beyond the effects of this austerity in many ways. For one, of the debt will create an opportunity for a new generation to get out of the burden of the debt. If a de-biz for all tax plan is publicly announced.Beyond that is however, self-governance will take, and the consequences mean be -- and is obvious. from here to America, as I wrote here, recently, debt

Figure 5: The generated text for NFE=16.

it's just me, it's bad feelings.He tells aware of sadistic behavior that is a joy to watch as a missionary and me when I see some right-handed people validating Mormonism to fall in love, and some I know are not, as I tell him. I teach me the left- hands, so does he. I love him for having his son. "He's often running riot," he says. "Like a kid bit in a get-out-of-way mode." I help him study a lot about that and I teach him about them.As a missionary I'm all out in search of sacred books, literally and figuratively. Every. I help him learn and I learn of in the way we prepare for mission. Again, these are complements of his ministry. "Man is man and she is fairly simple too, because I can see the power of that saying. when I'm all over him to evangel I really struggle to deal with it.But, I feel grounded in my own life. Though I may know what to read, I study scriptures, read sacred books and now when your eyes see this as clearly as I do, that counts as a difference. that you feel he is lonely. It, beautiful and is that, when you never, you realize he is brokenhearted. But from the best of his, it's just a beautiful reflection of his desire to experience in his daily life and the desires and needs of those he condemns and tries to project them onto young people.Either way, I welcome it.He is a human being and—I love him just as well as I can. I will not deny his character as a kind, humble, and hospitable man who loves us in many different ways, tender-hearted and will be some the greatest on the feet of the earth.He loves us all, but as he discerns more and more about people.That said, I'll love him a part of his missionary call. He knows I shouldn't change my mind and when he has come to be the Savior, every response creates rays of change in my heart.He loves us about people every day in the work of God, through his mission to "correct the whole world." The better he can, the more he can come to understand people.Sometimes, he says that he casts a shadow for th

Figure 6: The generated text for NFE=32.

we're going to let the players know that some new players have a responsibility to the fans.Q: Do you have a idea of it like this, how to the teams with the fans?No, before that, went into this season, we were at the moment we were heading for the playoffs in this game.The only difference is that, at the beginning, we had seven to go. in that moment, you put on the match with the players that popped up from the field with the players only, which is hard to do, because there are so many to deal with that we had to mess around with. And there, it makes it over to the training room"the training room.I think everyone on the team is in the training room.In the training room, the players come out here, the players come to the standing on the ground, the pass the ball.When the head of the player passes, the player has to push-up their head on the ground, if it from the pass. thenall the players, who are going to pass when the touch the ball, have to push-up their heads on this when they receive the ball before they receive the ball.The organization knows that the match is not won not just by the fans' support, so that first half the fans' fans decided to become fans in the training room, where the players can help provide the input by looking at the defender's tips.The defense is going to score the goal's at the match's end. That will be based on the first half. This is when we know it's going to be difficult for them.The team's title is going to be for the people. That's the title, for the people.Q: That's the talk that you gave the players this past week for in Tuesday. A number of times they didn't really set up routines for the game.Yes. They didn't know. I mean: I don't know, in the team group, I just think that they all the whatever it is, they just got to prepare themselves for the easiest any time, and when we play the game for a week, then we'll go to something to the rest.For the players, we are the ones to have the preparation routine, the ones to

Figure 7: The generated text for NFE=64.

