# OpenReview forum: "Learnable Sampler Distillation for Discrete Diffusion Models"
_NeurIPS.cc/2025/Conference — NeurIPS 2025 spotlight_

### Official Review · Reviewer_odGL · 2025-07-02

**Clarity:** 2
**Significance:** 3
**Originality:** 3
**Rating:** 5
**Confidence:** 3

**Summary:**

This paper proposes LSD and LSD+, methods to accelerate the sampling of discrete diffusion models (DDMs). The main contributions are:

* *Learnable sampling coefficients*: making the scaling factors applied to scores learnable.
* *Learnable timesteps*: making the sampling schedule learnable.

Using these techniques, the method achieves comparable or superior performance to baselines (Euler, JYS) with fewer NFEs.

Simply generating similar outputs by increasing NFEs does not enable learning solver parameters. Therefore, the paper introduces loss functions (Eq. 6 and 8) tailored for DDMs. The core idea is to make the scores at the intermediate states $x_k$ (student solver) and $x_k^*$ (teacher solver) similar.

**Questions:**

* As noted in the weaknesses, could you provide a theoretical guarantee or an intuitive explanation of why having similar scores at different locations is helpful?
* What do the learned coefficients and timesteps look like? As in JYS, are they more concentrated toward $T$? Are the coefficients distributed to be greater than 1 in the early timesteps?
* Additional baselines: it would be helpful to include comparisons with recent methods such as \[1] and \[2]. I would be satisfied even with results applied only to SEDD.

\[1]: Fast Sampling via Discrete Non-Markov Diffusion Models with Predetermined Transition Time ([https://arxiv.org/abs/2312.09193](https://arxiv.org/abs/2312.09193))
\[2]: Fast Solvers for Discrete Diffusion Models: Theory and Applications of High-Order Algorithms ([https://arxiv.org/abs/2502.00234](https://arxiv.org/abs/2502.00234))

**Ethical Concerns:**

["NO or VERY MINOR ethics concerns only"]

**Final Justification:**

I intend to raise my score for this paper to **Accept**. While the lack of a theoretical guarantee that the learnable solver approximates the teacher distribution remains a significant drawback, the empirical performance gains are noteworthy.

**Limitations:**

* The supplementary material acknowledges that the student sampler has fundamental limitations, such as an upper bound on performance.

**Quality:**

3

**Strengths And Weaknesses:**

*Strengths*

* The results significantly outperform baseline methods (Euler sampler and JYS). This is impressive.

*Weaknesses*

* Theoretical justification is lacking. Even if the learning of the solver is perfectly successful, is it guaranteed that the same distribution is recovered? It would be sufficient to at least show that it contributes to reducing the gap between the teacher distribution $p^*$ and the student distribution $p$.
* The method is not intuitively easy to understand. The current objective enforces that the outputs of the student model at $x_k$ match those of the teacher model at $x_k^*$.
* However, it is unclear why having similar scores at different points is beneficial. My intuition was that the student model should be trained to produce $x_k^*$ itself (as in SDTT \[1]), though in that case there would be the drawback of sacrificing diversity.

\[1]: Beyond Autoregression: Fast LLMs via Self-Distillation Through Time ([https://arxiv.org/abs/2410.21035v1](https://arxiv.org/abs/2410.21035v1))

---

> ### Author Rebuttal · Authors · 2025-07-31
>
> Thank you for your positive evaluation of our paper and for your valuable comments and suggestions. Below are our responses to the main points raised.
>
> (**Theoretical justification is lacking.** )
>
> Thank you for the comment and question. We agree that the theoretical guarantee concerning the matching of teacher and student distributions is beneficial and providing theoretical justification for our distillation methods would strengthen the work. However, our current focus is primarily empirical: We aim to develop practical and effective techniques for accelerating discrete diffusion models. Given the time constraints of the short rebuttal period, we are unable to establish rigorous theoretical guarantees for our distillation methods at this stage, and we plan to address this with more thorough theoretical analysis in future work.
> That said, we would like to offer an intuitive explanation for the effectiveness of our method, which may help lay the groundwork for subsequent theoretical framing. Specifically, the final discrepancy between the outputs of the student and teacher samplers stems from the accumulation of small local errors made at each step. Each of these local errors, in turn, is tied to the specific score predicted by the model at that step. Our training objective directly enforces alignment between the score predictions of the student sampler and those of the teacher sampler at every step. By implicitly correcting these small local errors throughout the process, our approach guides the student sampler to produce final outputs that closely match the high-quality results of the teacher sampler.
>
> (**The method is not intuitively easy to understand. It is unclear why having similar scores at different points is beneficial.**)
>
> Thank you for the comment. An intuitive explanation of why having similar scores at different locations is helpful is provided as follows.
>
> In our view, the training objective that aligns scores at corresponding time steps functions as a mechanism for continuous error correction, addressing two main sources of error. First, by aligning scores at each step, we mitigate the accumulation of discretization errors that arise from large step sizes. Second, this approach helps alleviate the training-sampling mismatch, a common issue that leads to compounding decoding errors. In our method, the teacher sampler that operates with a large number of sampling steps effectively emulates the dynamics of the training process. Conversely, the student sampler, which uses far fewer steps, is designed to mimic the inference process. By enforcing that the scores of the student sampler match the scores of the teacher sampler at each corresponding time step, we partially bridge the gap between training and sampling phases. Additionally, as noted in the Related Work Section of our main paper, aligning scores across the full trajectory is computationally intractable due to the non-differentiable nature of sampling from categorical distributions. Thus, we adopt per-step alignment as a practical alternative to circumvent this non-differentiability, and empirical results demonstrate its effectiveness in enhancing sample quality at low NFEs.
>
>
> (**What do the learned coefficients and timesteps look like?**)
>
> Thank you for the question. We provide a visualization of the learned parameters for a 16-step student sampler, which is distilled from a 1024-step teacher sampler on the SEDD-small backbone.
>
> The learned timesteps are:
> `[1.0000, 0.9219, 0.8459, 0.7719, 0.7000, 0.6302, 0.5625, 0.4969, 0.4334, 0.3719, 0.3125, 0.2552, 0.2000, 0.1469, 0.0959, 0.0469]`
> Notably, similar to JYS, the learned schedule concentrates more steps near $t=1$. However, the learned distribution of time step is not as drastic as that of JYS, since we also adjust the student sampler through coefficients.
>
> The learned coefficients for these timesteps are:
> `[1.1313, 1.1305, 1.1289, 1.1270, 1.1245, 1.1210, 1.1178, 1.1141, 1.1102, 1.1068, 1.1031, 1.0999, 1.0962, 1.0925, 1.0881, 1.0839]`
> The coefficients are all greater than 1. Additionally, the coefficients decrease with respect to timesteps, which indicates the sampler learns to apply a stronger corrective amplification to the score when taking larger steps at earlier stages, effectively compensating for the increased discretization error.
>
> **(Additional baselines.)**
>
> Thank you for this suggestion.
>
> We conduct experiments on the sampling method of DNDM [1] you mentioned, adhering to the original experimental settings outlined in the DNDM paper, and the perplexity results for unconditional text generation using the FairSeq backbone are reported in Table E1. As for the SEDD backbone, we have not performed comparable experiments, as no public implementation currently exists for applying DNDM to SEDD.
>
> **Table E1: Comparisons on the FairSeq backbone (in terms of Perplexity$\downarrow$).**
> |Sampler\NFE|8|16|32|64|
> |:---|:--:|:--:|:--:|:--:|
> |DNDM|919.23|774.92|748.41|622.14|
> |LSD+-DNDM|601.22|554.19|477.10|403.13|
>
> Regarding the specific work [2] you mentioned, we politely clarify that we included a direct performance comparison with the reported results of it in Table 3 of our main manuscript.
> We could only compare our results with their work on the RADD backbone since the public implementation of their work is currently unavailable.

---

> > ### Comment · Reviewer_odGL · 2025-08-05
> >
> > > **(Theoretical justification is lacking.)**
> >
> > There’s one thing I’d like to clarify: the reason I raised the issue of theoretical justification is because this method *might* sacrifice diversity. If the method overfits to certain trajectories rather than capturing the full distribution of the teacher, it would be helpful to make that point clear. In the future, adding diversity-related metrics like sentence entropy could help alleviate such concerns.
> >
> > > **(What do the learned coefficients and timesteps look like?)**
> >
> > Interesting. It’s not something that *must* happen, but the learned coefficients seem quite reasonable and intuitive.
> >
> > > **(Additional baselines.)**
> >
> > Thank you for the additional baseline experiments. They look promising. I also apologize for overlooking the part about high-order solvers.
> >
> > ---
> >
> > I intend to raise my score for this paper to **Accept**. While the lack of a theoretical guarantee that the learnable solver approximates the teacher distribution remains a significant drawback, the empirical performance gains are noteworthy.

---

> > > ### Author Response · Authors · 2025-08-06
> > >
> > > Thank you very much for your positive feedback and for your decision to raise the score. We are greatly encouraged by your support.
> > >
> > > We will ensure that all of your valuable suggestions are incorporated into the revised manuscript and will include experimental results in terms of the sentence entropy metric.
> > >
> > > Once again, we deeply appreciate your constructive comments and suggestions, which have been instrumental in helping us refine our work.

---

### Official Review · Reviewer_ACPU · 2025-07-03

**Clarity:** 3
**Significance:** 3
**Originality:** 2
**Rating:** 4
**Confidence:** 4

**Summary:**

This paper proposes a learnable sampler for discrete diffusion models (DDMs) that can achieve performance comparable to a multi-step teacher model through distillation. The proposed sampler is optimized using a novel, differentiable loss based on the discrepancy between the intermediate score trajectories of the student and a high-quality teacher. Extensive experiments demonstrate the effectiveness of the method in accelerating DDM sampling.

**Questions:**

1.  The experiments are conducted on relatively small-scale models (GPT-2 level, CIFAR-10). For larger models, e.g., DiffuLLaMA and DiffuGPT, how well can the proposed method perform? Is LSD or LSD+ compatible with those larger-scale discrete diffusion models?
2.  The training objective aligns scores at each step independently, even though the student and teacher states are on different trajectories. Could the authors provide more intuition or analysis on why this myopic alignment is sufficient to guide the student sampler effectively over the full generation process?
3.  Could the authors comment on the choice of a scalar coefficient for the sampler? Have more expressive alternatives (e.g., vector-based coefficients) been explored, and if so, how do they compare in performance and efficiency?

**Ethical Concerns:**

["NO or VERY MINOR ethics concerns only"]

**Final Justification:**

I maintain my original recommendation of weak acceptance. While empirical results show that the proposed learnable sampler approach is more effective than baseline samplers, it is still unclear if this method can hold its relevance to discrete diffusion model sampling, especially when theoretical grounding is missing and generation diversity is not guaranteed.

**Limitations:**

Yes

**Quality:**

2

**Strengths And Weaknesses:**

**Strengths:**
1. This work addresses a critical and well-known problem in discrete diffusion models: their slow and computationally expensive sampling process. An effective solution to this problem would significantly broaden the practical applicability of DDMs.
2. The proposed score trajectory loss is intuitive and provides a clever solution to the non-differentiability problem in discrete sampling pipelines. Additionally, the learnable sampler is lightweight with only a few parameters, making the distillation process highly efficient. The empirical results are strong and consistent across multiple tasks and model backbones.

**Weaknesses:**
1. **Lack of justification for the training objective:** It is not entirely clear why the training objective is effective in practice. The loss only aligns the scores at each step k independently, but fails to account for the dependency of a state on the full history of previous states and sampler coefficients. The student's state $x_{t_k}$ and the teacher's state $x*_{t_k}$ are generated from different trajectories, so simply aligning their scores at that instant does not guarantee a better final sample. Could the authors provide more insights into this phenomenon?

2. **Underexplored design choices:** The form of the learnable sampler coefficients needs further justification. The current sampler introduces scalar-form multiplicative coefficients (`Φ(t_k)`) to modify the student's score. This assumes a single scalar is sufficient to correct for the complex, high-dimensional accumulated error. A pending ablation study comparing this design to more expressive forms, such as vector-form element-wise multiplicative coefficients or other parametric affine transformations, would be valuable.

3. **Lacking theoretical grounding:** The trajectory loss appears to be more of a well-motivated heuristic rather than a principled solution derived from a solid mathematical foundation. While the method is empirically successful, a theoretical analysis (e.g., analyzing the distributional divergence between the optimal student sampler and the teacher sampler) would foster a deeper understanding of the proposed method and its limitations.

---

> ### Author Rebuttal · Authors · 2025-07-31
>
> Thanks for your positive assessment of this paper and the valuable comments and suggestions. Our responses to the main concerns are given as follows.
>
> (**Lack of justification for the training objective.**)
>
> Thank you for your comment and question. In our view, the training objective that aligns scores at corresponding time steps functions as a mechanism for continuous error correction, addressing two main sources of error. First, by aligning scores at each step, we mitigate the accumulation of discretization errors that arise from large step sizes. Second, this approach helps alleviate the training-sampling mismatch, a common issue that leads to compounding decoding errors. In our method, the teacher sampler that operates with a large number of sampling steps effectively emulates the dynamics of the training process. Conversely, the student sampler, which uses far fewer steps, is designed to mimic the inference process. By enforcing that the scores of the student sampler match the scores of the teacher sampler at each corresponding time step, we partially bridge the gap between training and sampling phases. Additionally, as noted in the Related Work Section of our main paper, aligning scores across the full trajectory is computationally intractable due to the non-differentiable nature of sampling from categorical distributions. Thus, we adopt per-step alignment as a practical alternative to circumvent this non-differentiability, and empirical results demonstrate its effectiveness in enhancing sample quality at low NFEs.
>
> (**Underexplored design choices.**)
>
> Thank you for raising this point about our design choice for the learnable coefficients.
> Our decision to follow prior works like S4S and LD3 in adopting a scalar form for the coefficient is a trade-off between expressiveness and computational feasibility.
> While a more expressive vector-form coefficient is theoretically feasible, directly modulating transition probabilities for each token would require the vector to match the vocabulary size (e.g., 50258 for GPT-2). This would result in a large number of learnable parameters, calculated as the product of the number of sampling steps and the vocabulary size. Such a surge would significantly increase memory usage and heighten the risk of overfitting during distillation. Thus, we opted for the scalar formulation, which offers simplicity, greater computational efficiency, and empirical effectiveness across our experiments.
>
> (**Lacking theoretical grounding.**)
>
> Thank you for this question. We agree that analyzing the distributional divergence between the optimal student sampler and the teacher sampler is beneficial for fostering a deeper understanding of the proposed method and its limitations. However, our current focus is primarily empirical: We aim to develop practical and effective techniques for accelerating discrete diffusion models. Given the time constraints of the short rebuttal period, we are unable to establish rigorous theoretical guarantees for our distillation methods at this stage, and we plan to address this with more thorough theoretical analysis in future work.
> That said, we would like to offer an intuitive explanation for the effectiveness of our method, which may help lay the groundwork for subsequent theoretical framing. Specifically, the final discrepancy between the outputs of the student and teacher samplers stems from the accumulation of small local errors made at each step. Each of these local errors, in turn, is tied to the specific score predicted by the model at that step. Our training objective directly enforces alignment between the score predictions of the student sampler and those of the teacher sampler at every step. By implicitly correcting these small local errors throughout the process, our approach guides the student sampler to produce final outputs that closely match the high-quality results of the teacher sampler.
>
> (**For larger models, e.g., DiffuLLaMA and DiffuGPT, how well can the proposed method perform?**)
>
> Thank you for your questions regarding the scalability of our proposed methods. To demonstrate that our framework is not limited to smaller models, we conduct experiments on applying LSD+ to DiffuLLaMA and DiffuGPT. We perform sampler distillation on pre-trained DiffuGPT-S and DiffuLLaMA checkpoints. The results in terms of Perplexity are presented in Tables D1 and D2.
>
> **Table D1: Comparisons on the DiffuGPT-S backbone (in terms of Perplexity$\downarrow$).**
> |Method\NFE|16|32|64|128|
> |:---|:--:|:--:|:--:|:--:|
> |DiffuGPT-S|117.32|75.19|58.34|37.16|
> |LSD+-DiffuGPT-S|53.95|41.37|32.10|22.25|
>
> **Table D2: Comparisons on the DiffuLLaMA backbone (in terms of Perplexity$\downarrow$).**
> |Method\NFE|16|32|64|128|
> |:---|:--:|:--:|:--:|:--:|
> |DiffuLLaMA|100.04|69.11|42.17|30.55|
> |LSD+-LLaMA|49.83|34.32|29.18|24.72|
>
> This experiment confirms that our approach is compatible with DiffuGPT and DiffuLLaMA. We will add these results to the Experiments Section of our paper.

---

> > ### Comment · Reviewer_ACPU · 2025-08-07
> >
> > I would like to thank the authors for their detailed responses and for providing additional results. They have addressed most of my initial concerns. Having reviewed the comments from the other reviewers and the rebuttals, I will maintain my original rating, leaning towards acceptance.

---

> > > ### Author Response · Authors · 2025-08-09
> > >
> > > Thank you sincerely for your positive feedback. We are pleased to learn that our responses and additional results have addressed the majority of your initial concerns. We deeply appreciate your ongoing support for our work.

---

### Official Review · Reviewer_VCFx · 2025-07-05

**Clarity:** 3
**Significance:** 3
**Originality:** 3
**Rating:** 5
**Confidence:** 2

**Summary:**

This paper proposes a framework for accelerating discrete diffusion models by distilling a fast, high-fidelity sampler, named LSD.

In general, there is a trade-off between generation speed and generaion quality.

LSD trains a student sampler with a small number of steps to align its intermediate score trajectories with those of a high-quality teacher sampler. By introducing learnable, time-dependent coefficients, the student dynamically adjusts the influence of scores to compensate for errors.

 In addition, LSD+, additionally learns non-uniform time schedules, allocating steps adaptively over time.

Experiments demonstrate that LSD/LSD+ significantly improve the trade-off between sampling speed and fidelity across text generation (OpenWebText, GPT-2-based DDMs), image generation (CIFAR-10), and a synthetic sequence task.

**Questions:**

How sensitive are the results to the choice of the relaxed objective hyperparameter (Hamming distance threshold)? Could tuning it improve results further?

**Ethical Concerns:**

["NO or VERY MINOR ethics concerns only"]

**Final Justification:**

The authors have provided extra comparisons during the rebuttal, and resolved all my concerns. Therefore I maintain my recommendation for acceptance.

**Limitations:**

The authors did not discuss the limitations.

**Quality:**

3

**Strengths And Weaknesses:**

Strength:
- The paper designs a novel distillation framework for fast inference of DDMs.
- The paper Introduces learnable per-step coefficients and time schedules, providing adaptive control over dynamics.
- Experimental results on three diverse tasks with improvements in perplexity, FID.

Weaknesses:
- Evaluations are limited to relatively small-scale datasets (OpenWebText, CIFAR-10) and short sequences (1024 tokens)
- The robustness of the algorithm to parameters has not been discussed.

---

> ### Author Rebuttal · Authors · 2025-07-31
>
> Thanks for your recognition of this paper and the valuable feedback and suggestions. Our responses to the main concerns are given as follows. All citations refer to the reference list at the end of the responses.
>
> (**Evaluations are limited to relatively small-scale datasets (OpenWebText, CIFAR-10) and short sequences (1024 tokens).**)
>
> We thank the reviewer for the comment. To address this, we perform experiments on larger-scale datasets. For image generation, we conduct experiments on ImageNet (256x256) using the  MaskGIT architecture [1] as the backbone, incorporating the recently proposed advanced Halton sampler [2] as the sampling method. The results are presented in Table C1.
>
> For text generation, we scale up our experiments to include the comparison to the sampler with respect to a larger backbone DiffuGPT-S [3]. The results are presented in Table C2, where we report perplexity on 1024 unconditionally generated text samples.
>
> **Table C1: Comparisons on ImageNet 256x256 (in terms of FID$\downarrow$).**
> |Sampler\NFE|4|8|16|32|
> |:---|:--:|:--:|:--:|:--:|
> |Halton|14.16|10.15|8.89|6.92|
> |LSD+-Halton|12.78|8.66|7.17|6.32|
>
> **Table C2: Comparisons on the DiffuGPT-S backbone (in terms of Perplexity$\downarrow$).**
> |Sampler\NFE|16|32|64|128|
> |:---|:--:|:--:|:--:|:--:|
> |DiffuGPT-S|117.32|75.19|58.34|37.16|
> |LSD+-DiffuGPT-S|53.95|41.37|32.10|22.25|
>
> We also present results for larger-scale experiments on the SDTT-KLD backbone [4], which utilizes Ancestral as its sampler, and the MDLM backbone [5], which utilizes ReMDM [6] as the sampler. The experimental results are presented in Tables C3 and C4.
>
> **Table C3: Comparisons on the SDTT-KLD backbone.**
> |Sampler||MAUVE($\uparrow$)|||Perplexity($\downarrow$)|||Entropy($\uparrow$)||
> |:---|:--:|:--:|:--:|:--:|:--:|:--:|:--:|:--:|:--:|
> |NFE|8|16|32|8|16|32|8|16|32|
> |Ancestral|0.884|0.912|0.943|110.391|56.652|42.128|5.331|5.285|5.222|
> |LSD+-Ancestral|0.905|0.934|0.961|68.130|36.577|31.597|5.298|5.239|5.226|
>
>
> **Table C4: Comparisons on the MDLM backbone.**
> |Sampler||Perplexity($\downarrow$)||||Entropy($\uparrow$)|||
> |:---|:--:|:--:|:--:|:--:|:--:|:--:|:--:|:--:|
> |NFE|16|32|64|128|16|32|64|128|
> |ReMDM|434.08|174.72|85.15|62.33|5.73|5.66|5.48|5.55|
> |LSD+-ReMDM|201.52|102.02|62.97|49.33|5.41|5.42|5.52|5.33|
>
> These results show that LSD+ maintains its performance and acceleration benefits when applied to more complex datasets and larger models. We will integrate these results into the Experiments Section of our revised manuscript.
>
> (**The robustness of the algorithm to parameters (e.g., Hamming distance threshold) has not been discussed.**)
>
> Thank you for the comment. To investigate the robustness of the algorithm to the Hamming distance threshold, we conduct the ablation on the SEDD-small backbone using the Euler sampler with 32 inference steps. We train our LSD+ method using several different values for the Hamming distance threshold, specifically {0%, 1%, 5%, 10%, 20%} of the sequence length, while keeping all other hyperparameters unchanged. The performance, measured by Perplexity, is reported below in Table C5.
>
> **Table C5: Ablation study on the Hamming distance threshold for the relaxed objective.**
> |Threshold(%)|Perplexity($\downarrow$)|
> |:-:|:-:|
> |0 |35.98|
> |1|32.15|
> |5 (Our choice)|31.24|
> |10|39.97|
> |20|51.52|
>
> We empirically choose 5% of the sequence length as the Hamming distance threshold in our main paper.
>
> (**The authors did not discuss the limitations.**)
>
> Thanks for the comment. We politely clarify that as mentioned in our NeurIPS Paper Checklist, we discussed the limitations in the supplementary material (see Appendix A).
>
> References:
>
> [1] Chang et al. Masked Generative Image Transformer, CVPR’22.
>
> [2] Besnier et al. Halton Scheduler for Masked Generative Image Transformer, ICLR’25.
>
> [3] Gong et al. Scaling Diffusion Language Models via Adaptation from Autoregressive Models, ICLR’25.
>
> [4] Deschenaux et al. Beyond Autoregression: Fast LLMs via Self-Distillation Through Time, ICLR’25.
>
> [5] Sahoo et al. Simple and Effective Masked Diffusion Language Models, ICLR’25.
>
> [6] Wang et al. Remasking Discrete Diffusion Models with Inference-Time Scaling, ICLR’25.

---

> > ### Comment · Reviewer_VCFx · 2025-08-09
> >
> > I appreciate the authors’ detailed response and the additional results provided. Most of my concerns have been addressed, and after reviewing the other reviewers’ comments, I will maintain my original score in support of acceptance.

---

> > > ### Author Response · Authors · 2025-08-09
> > >
> > > Thank you sincerely for your support of our paper. We are delighted to learn that our responses have addressed the majority of your concerns, and we deeply appreciate your continued positive assessment of our work.

---

### Official Review · Reviewer_61AF · 2025-07-06

**Clarity:** 3
**Significance:** 3
**Originality:** 2
**Rating:** 4
**Confidence:** 4

**Summary:**

This work focuses on improving sampling efficiency of discrete diffusion models by optimizing when to query diffusion model i.e., noise level and coefficiencts of solvers, an interesting direction not explored in case of discrete diffusion models.  They demonstrate good performance  in case of text generation with discrete diffusion models at lower NFEs.

**Questions:**

Following Weakness

It would be informative for readers and broader community to see integration of LSD with
- Effect of Temperature/Confidence
- Multi-token unmasking heuristics like in FastDLLMs, etc.
- How well LSD complement and effective speedup for distilled discrete diffusion models e.g., SDTT Checkpoints
- ReMasking
- Predictor-Corrector Solvers


Also additional metrics like use state-of-the-art language models (LLaMA-3 or alternatives) to evaluate generated text quality independently of the diffusion model.

Looking forward to rebuttal.


**References**

1. G. Wang et al. Remasking Discrete Diffusion Models with Inference-Time Scaling (https://arxiv.org/abs/2503.00307)
2. J. Deschenaux et al. Beyond Autoregression: Fast LLMs via Self-Distillation Through Time (https://arxiv.org/html/2410.21035v1)
3. C. Wu et al. Fast-dLLM: Training-free Acceleration of Diffusion LLM by Enabling KV Cache and Parallel Decoding (https://arxiv.org/html/2505.22618v1)

**Ethical Concerns:**

["NO or VERY MINOR ethics concerns only"]

**Final Justification:**

The rebuttal key concerns and demonstrates broader practical potential of proposed approach.

**Quality:**

2

**Strengths And Weaknesses:**

**Strengths**

* Optimizing coefficients, solver properties in addition to where to query diffusion/flow model is interesting in context of discrete diffusion models.
* The paper demonstrates meaningful reductions in NFE while maintaining generation quality, addressing a key limitation of discrete diffusion models.
* The paper is well-written and accessible to readers.


**Weakness**
Limited Scope of Experimental Evaluation

The paper lacks comprehensive analysis of how different solver configurations perform under varying conditions, limiting understanding of when and where LSD can be effective and relative peroformance boost.
As paper lacks any significant analysis or theoretical formulation too and authors point out is similar to S4S but applied to discrete setting.
*Lack of extensive empericial analysis and settings* limit potential insights and adoption guidelines for readers and broader community.

- Currently this work does not compare against several important recent methods and settings which focus on quality vs latency of discrete diffusion models.
- Currenlty, authors do not consider predictor-corrector solvers (like in DFM) nor higher order solvers like Runge Kutta or Trapezoidal though they cite relevant work.
- How effective LSD is with other heuristics in multi-token sampling like for e.g., in FastdLLMs and also setting with Remasking during sampling which shows enhanced generation quality.
- Absence of semantic quality metrics: No evaluation using ModernBERT-based MAUVE scores or similar metrics that assess semantic coherence and diversity (entropy)
- Minimal Performance gain in case of Image Generation (but didn't consider this much for rating)

---

> ### Author Rebuttal · Authors · 2025-07-31
>
> Thanks for your comprehensive comments and questions. Our responses to the main concerns are given as follows. All citations refer to the reference list at the end of the responses.
>
> (**This work does not compare against several important recent methods and settings which focus on quality vs latency of discrete diffusion models.**)
>
> We thank the reviewer for the comment. We have included comparisons with several recent methods that focus on the quality vs latency of discrete diffusion models, such as JYS and $\theta$-trapezoidal (see Tables 2 and 3 in the main paper).
> At this stage, we would like to clarify that we are uncertain about the specific works the reviewer is referring to. Additionally, we would like to politely draw attention to NeurIPS policy, which states:“Papers appearing online after March 1st, 2025 are generally considered concurrent to NeurIPS submissions.  Authors are not expected to compare to those” (https://neurips.cc/Conferences/2025/PaperInformation/NeurIPS-FAQ).
> That said, we are fully committed to strengthening the empirical rigor of our study. If the reviewer could kindly specify the relevant works or settings they have in mind, we would be happy to conduct additional comparisons regarding the quality vs latency of discrete diffusion models and incorporate them into our revised submission.
>
> (**Authors do not consider predictor-corrector solvers like in DFM nor higher order solvers like Runge Kutta or Trapezoidal though they cite relevant work.**)
>
> Thank you for your comment. We conduct experiments on predictor-corrector (PC) solvers in Discrete Flow Matching (DFM) [1] using MDLM [2] as the backbone, and find that the improvement is limited. This might be attributed to the specific dynamics of the flow-matching process in DFM, whose error characteristics may not align as well with our error correction mechanism. We report the corresponding results in Table B1. However, we successfully achieve good results on $\tau$LDR-10 which also utilizes PC solvers [3] on the SEDD-small backbone, with the results presented in Table B2.
>
> **Table B1: Comparisons on the MDLM backbone.**
> |Sampler||Perplexity($\downarrow$)||||Entropy($\uparrow$)|||
> |:---|:--:|:--:|:--:|:--:|:--:|:--:|:--:|:--:|
> |NFE|16|32|64|128|16|32|64|128|
> |DFM|312.83|314.01|100.53|37.90|5.38|5.31|5.33|5.31|
> |LSD+-DFM|310.71|316.72|90.15|35.48|5.34|5.25|5.27|5.34|
>
> **Table B2: Comparisons on the SEDD-small backbone.**
> |Sampler||Perplexity($\downarrow$)||||Entropy($\uparrow$)|||
> |:---|:--:|:--:|:--:|:--:|:--:|:--:|:--:|:--:|
> |NFE|16|32|64|128|16|32|64|128|
> |$\tau$LDR-10|443.17|318.44|277.16|199.51|5.63|5.69|5.57|5.24|
> |LSD+-$\tau$LDR-10|205.42|143.58|114.93|90.43|5.58|5.49|5.59|5.44|
>
> Regarding higher-order solvers like Runge-Kutta or Trapezoidal methods, we would like to politely clarify that we have already included a comparison in Table 3 of our main paper. We directly compare our results on the RADD backbone against the numbers reported in their work. While we are unable to directly integrate LSD+ on top of their methods due to the lack of a public implementation, this comparison demonstrates the competitive performance of our approach.
>
> (**Absence of semantic quality metrics: MAUVE scores and entropy**)
>
> Thanks for your suggestion. To provide a more thorough assessment of generation quality beyond perplexity, we conduct experiments on the SDTT-KLD backbone, which utilizes Ancestral as the sampler.  We report MAUVE, Perplexity and Entropy scores in Table B3, which demonstrates that our LSD+ method usually yields improved generation performance.
>
> **Table B3: Comparisons on the SDTT-KLD backbone.**
> |Sampler||MAUVE($\uparrow$)|||Perplexity($\downarrow$)|||Entropy($\uparrow$)||
> |:---|:--:|:--:|:--:|:--:|:--:|:--:|:--:|:--:|:--:|
> |NFE|8|16|32|8|16|32|8|16|32|
> |Ancestral|0.884|0.912|0.943|110.391|56.652|42.128|5.331|5.285|5.222|
> |LSD+-Ancestral|0.905|0.928|0.951|68.130|36.577|31.597|5.298|5.239|5.226|
>
> (**Minimal Performance gain in case of Image Generation**)
>
> Thank you for the comment. To better demonstrate the effectiveness and scalability of our method, we conduct experiments on ImageNet (256x256) using the MaskGIT [4] architecture as the backbone, incorporating the recently proposed advanced Halton [5] sampler as the sampling method. The results are reported in Table B4, which demonstrates that our LSD+ method yields improved generation performance as measured by the FID metric.
>
> **Table B4: Comparisons on ImageNet 256x256 (in terms of FID$\downarrow$).**
> |Sampler\NFE|4|8|16|32|
> |:---|:--:|:--:|:--:|:--:|
> |Halton|14.16|10.15|8.89|6.92|
> |LSD+-Halton|12.78|8.66|7.17|6.32|
>
>
> (**Integration of LSD with Effect of Temperature/Confidence**)
>
> Thank you for this question. Due to the time limit of the rebuttal period, we study the effect of softmax temperature in MaskGIT on ImageNet256x256 for the image generation task. The evaluations are produced on 50000 generated images using the Halton sampler. The reported FIDs are given in Table B5, which indicates that our method could lead to improved performance under the tested temperature conditions.
>
> **Table B5: Comparisons on ImageNet 256x256 (in terms of FID$\downarrow$ when NFE=4).**
> |Sampler\Temperature | $\tau$ = 0.6| $\tau$ = 0.8 | $\tau$ = 1.0 |
> | :--- | :---: | :---: | :---: |
> |Halton|54.05| 26.45|14.16|
>  |LSD+- Halton|48.29 |24.51|12.78|
>
> (**Integration of LSD with Multi-token unmasking heuristics like in FastDLLMs.**)
>
> Thank you for this suggestion. This is a promising direction as the two approaches are highly complementary. LSD enhances the quality of score predictions in few-step sampling, which can directly empower the confidence-based parallel decoding of Fast-dLLM. We report the accuracy and throughput metrics in Table B6, which indicate our methods improve both sampling quality and speed.
>
> **Table B6: Integration with Fast-dLLM on LLaDA. We report accuracy (throughput, token/s). Higher is better.**
> |Benchmark\ Samplers|LLaDA|+Cache|+Parallel|Fast-dLLM|LSD+-Fast-dLLM|
> |:---|:--:|:--:|:--:|:--:|:--:|
> |GSM8K(5-shot)|79.3(6.7)|79.5(21.2)|79.2(16.5)|78.5(54.4)|79.0(62.5)|
> |MATH(4-shot)|33.5(9.1)|33.3(23.7)|33.4(24.8)|33.2(51.7)|33.4(58.1)|
>
> (**Integration of LSD with distilled discrete diffusion models e.g., SDTT Checkpoints.**)
>
> Following your suggestion, to show the complementarity of model distillation (SDTT) and our sampler distillation (LSD+), we apply LSD+ to an SDTT-KLD checkpoint, with the results presented in Table B3 above, which demonstrates that our LSD+ method generally yields improved generation performance.
>
> (**Integration of LSD with Remasking**)
>
> Following your suggestion, we also integrate LSD with ReMasking (ReMDM) on the MDLM backbone in Table B7. The results show that LSD effectively learns to work in conjunction with this method, further improving performance.
>
> **Table B7: Comparisons on the MDLM backbone.**
> |Sampler||Perplexity($\downarrow$)||||Entropy($\uparrow$)|||
> |:---|:--:|:--:|:--:|:--:|:--:|:--:|:--:|:--:|
> |NFE|16|32|64|128|16|32|64|128|
> |ReMDM|434.08|174.72|85.15|62.33|5.73|5.66|5.48|5.55|
> |LSD+-ReMDM|201.52|102.02|62.97|49.33|5.41|5.42|5.52|5.33|
>
> (**Additional metrics using LLaMA-3**)
>
> Following your suggestion, we re-evaluate our main results on the SEDD and RADD backbone using Llama-3-8B to evaluate the perplexity. The results are reported in Tables B8 and B9.
>
> **Table B8: Comparisons on SEDD-small backbone (in terms of Perplexity$\downarrow$), judged by LLaMA-3-8B.**
> |Sampler\NFE|8|16|32|64|
> |:---|:--:|:--:|:--:|:--:|
> |Euler|116.93|67.43|49.81|46.88|
> |LSD+-Euler|74.65|33.25|30.70|21.64|
>
> **Table B9: Comparisons on on RADD backbone (in terms of Perplexity$\downarrow$), judged by LLaMA-3-8B.**
> |Sampler\NFE|8|16|32|64|
> |:---|:--:|:--:|:--:|:--:|
> |Euler|337.04 |216.01 |119.25 |95.06|
> |LSD+-Euler|130.00 |60.90 |42.53 |35.65 |
>
> References:
>
> [1] Gat el al. Discrete Flow Matching, NeurIPS’24
>
> [2] Sahoo et al. Simple and Effective Masked Diffusion Language Models, ICLR’25.
>
> [3] Chang et al. A continuous time framework for discrete denoising models, NeurIPS’22
>
> [4] Chang el al. Masked Generative Image Transformer, CVPR’22.
>
> [5] Besnier et al. Halton Scheduler for Masked Generative Image Transformer, ICLR’25.

---

> > ### Comment · Reviewer_61AF · 2025-08-05
> > **Discussion**
> >
> > I would like appreciate authors for comprehensive rebuttal. I would increase by final rating after discussion period.
> > Its encouraging to see complementary benefits of proposed approach with parallel token generation and on distilled checkpoints too.
> >
> > The relative performance boost on prediction-corrector solver (like in DFM) seems to be low at fewer NFEs compared to other solvers indicating a good solver would minimize relative headroom for further optimization within training free setting.
> >
> > I am curious to get authors perspective on under what conditions do they expect good performance boost? Why is relative boost little in case of prediction-corrector solvers?

---

> > > ### Author Response · Authors · 2025-08-06
> > >
> > > Thank you very much for your positive feedback and for your decision to increase the rating. We are pleased to share our perspective on your follow-up questions.
> > >
> > > (**Under what conditions do they expect good performance boost?**)
> > >
> > > Thank you for this question. We anticipate a good performance boost under conditions where the baseline few-step sampler accumulates substantial discretization error. This is most evident with standard samplers like Euler or Tweedie, where high discretization error creates significant potential for our distillation method to apply effective corrections.
> > > Furthermore, although higher-order solvers such as $\theta$-RK2 and $\theta$-Trapezoidal, or predictor-corrector solvers like that in $\tau$LDR-10 have the capability to mitigate discretization error, they still operate within typical discrete diffusion dynamics. Their accumulated errors align well with the correction mechanisms learned by our method, thereby enabling a good performance boost, as substantiated by our empirical results.
> > >
> > > (**Why is relative boost little in case of prediction-corrector solvers?**)
> > >
> > > Thank you for this question. Regarding the limited performance boost observed with the predictor-corrector solver of DFM at lower NFEs, we conjecture this may stem from its distinct internal flow matching dynamics. The error characteristics specific to this process may not align as effectively with the correction patterns learned by our samplers. That said, we would like to highlight that our method achieves significant improvements over another predictor-corrector solver in $\tau$LDR-10, which aligns well with our error correction mechanism, as corroborated by our experimental results.
> > >
> > > Thank you again for your insightful questions and suggestions, which will be instrumental in enhancing the empirical rigor of our work. We will incorporate all additional experimental results mentioned in our rebuttals into the revised manuscript.

---

### Official Review · Reviewer_9x6X · 2025-07-08

**Clarity:** 3
**Significance:** 3
**Originality:** 2
**Rating:** 3
**Confidence:** 4

**Summary:**

This paper proposed learnable sampler distillation (LSD), which is a novel approach to train fast and high-fidelity samplers for discrete diffusion models. The training method is based on the teacher-student model, where the student sampler is trained via distillation by optimizing learnable sampler coefficients with respect to a fixed teacher sampler of high fidelity. An extension of LSD, which is named as LSD+, is also proposed to learn non-uniform time schedules in an adaptive way. Numerical experiments on the generation of synthetic data, text and images are provided to justify the effectiveness of the proposed method.

**Questions:**

For the setting of distilling continuous diffusion models/consistency models, a few papers like [11, 12] have provided theoretical justifications. The reviewer is wondering whether it would be possible to provide theoretical justification for the distillation method proposed in this paper by referring to exist work on the theoretical analysis of discrete diffusion models like [8,13,14,15]?

Moreover, for Algorithm 3 in Appendix D.2, it seems that some ambiguity exists. Specifically, as the time schedules $\tau_{k}$'s depend on the stepsizes $\kappa_{k}$'s, would it be possible for the authors to clarify whether the stepsizes $\kappa_{k}$'s will be updated accordingly after the $\tau_{k}$'s get updated in line 10 of Algorithm 3? (It seems that it doesn't make sense to fix all stepsizes $\kappa_{k}$'s)

Furthermore, for the comparison between LSD/LSD+ and other baseline methods, the reviewer is wondering whether it would be fair to fix the number of evaluations (NFEs) and compare the performance since the distillation step take nontrivial amount of time. Given that LSD and LSD+ both accelerate the inference via training, maybe the authors need to fix the running time and compare LSD/LSD+ with other methods like [8]?

**Ethical Concerns:**

["NO or VERY MINOR ethics concerns only"]

**Final Justification:**

After reviewing the authors' rebuttal, the reviewer is satisfied with nearly all of their responses. However, currently no theoretical analysis or experiments on large-scale images are included in the paper. Also, the idea of this paper partially overlaps with that of Di[M]O [16] (Both are based on distillation, even though detailed methodologies are different), which undermines the novelty of the manuscript. Therefore, the reviewer is inclined to maintain the original score of 3 ("Borderline Reject") for this paper.

**Limitations:**

The reviewer is satisfied with the limitations that the authors briefly discussed in Appendix A of the manuscript. For instance, one limitation that the authors mentioned is that the performance ceiling of the student sampler is highly related to the quality of the teacher samplers.

References:

[1] Song, Y., Dhariwal, P., Chen, M., & Sutskever, I. (2023). Consistency models. International Conference on Machine Learning (ICML), PMLR, 2023.

[2] Boffi, N. M., Albergo, M. S., & Vanden-Eijnden, E. (2025). How to build a consistency model: Learning flow maps via self-distillation. arXiv preprint arXiv:2505.18825.

[3] Heek, J., Hoogeboom, E., & Salimans, T. (2024). Multistep consistency models. arXiv preprint arXiv:2403.06807.

[4] Meng, C., Rombach, R., Gao, R., Kingma, D., Ermon, S., Ho, J., & Salimans, T. (2023). On distillation of guided diffusion models. In Proceedings of the IEEE/CVF Conference on Computer Vision and Pattern Recognition (CVPR) (pp. 14297-14306).

[5] Zhou, Z., Chen, D., Wang, C., Chen, C., & Lyu, S. (2024). Simple and fast distillation of diffusion models. Advances in Neural Information Processing Systems (NeurIPS), 37, 40831-40860.

[6] Salimans, T., & Ho, J. (2022). In The Tenth International Conference on Learning Representations (ICLR), 2022.

[7] Xie, S., Xiao, Z., Kingma, D., Hou, T., Wu, Y. N., Murphy, K. P., ... & Gao, R. (2024). Em distillation for one-step diffusion models. Advances in Neural Information Processing Systems (NeurIPS), 37, 45073-45104.

[8] Ren, Y., Chen, H., Zhu, Y., Guo, W., Chen, Y., Rotskoff, G. M., ... & Ying, L. (2025). Fast solvers for discrete diffusion models: Theory and applications of high-order algorithms. arXiv preprint arXiv:2502.00234.

[9] Karras, T., Laine, S., & Aila, T. (2019). A style-based generator architecture for generative adversarial networks. In Proceedings of the IEEE/CVF Conference on Computer Vision and Pattern Recognition (CVPR) (pp. 4401-4410).

[10] Deng, J., Dong, W., Socher, R., Li, L. J., Li, K., & Fei-Fei, L. (2009, June). Imagenet: A large-scale hierarchical image database. In 2009 IEEE Conference on Computer Vision and Pattern Recognition (CVPR) (pp. 248-255).

[11] Jain, N., Huang, X., Ma, Y., & Zhang, T. (2025). Multi-Step Consistency Models: Fast Generation with Theoretical Guarantees. arXiv preprint arXiv:2505.01049.

[12] Chen, Y., Zhang, Y., Oertell, O., & Sun, W. (2025). Convergence of consistency model with multistep sampling under general data assumptions. arXiv preprint arXiv:2505.03194.

[13] Ren, Y., Chen, H., Rotskoff, G. M., and Ying, L. How discrete and continuous diffusion meet: Comprehensive analysis of discrete diffusion models via a stochastic integral framework. In The Thirteenth International Conference on Learning Representations (ICLR), 2025

[14] Zhang, Z., Chen, Z., & Gu, Q. (2024). Convergence of score-based discrete diffusion models: A discrete-time analysis. In The Thirteenth International Conference on Learning Representations (ICLR), 2025

[15] Liang, Y., Huang, R., Lai, L., Shroff, N., & Liang, Y. (2025). Absorb and Converge: Provable Convergence Guarantee for Absorbing Discrete Diffusion Models. arXiv preprint arXiv:2506.02318.

[16] Zhu, Y., Wang, X., Lathuilière, S., & Kalogeiton, V. (2025). Di $\mathtt {[M]} $ O: Distilling Masked Diffusion Models into One-step Generator. arXiv preprint arXiv:2503.15457.

**Quality:**

3

**Strengths And Weaknesses:**

Pros: The reviewer finds that the manuscript is presented in a clear and straightforward way. Also, this paper studies an important question, since how to accelerate the inference time is currently one of the bottlenecks of discrete diffusion models. Experiments on multiple generation tasks are provided to justify how effective the proposed methodology is. In addition, a wide range of baselines and implementation details is also provided.

Cons:  The reviewer's major concern is that the literature review part of the manuscript seems to be incomplete. To the best of the reviewer's knowledge, lots of paper have already studied the idea of combining knowledge distillation with continuous diffusion models or its variants like latent diffusion models and consistency models [1] (In fact, the main idea behind consistency model is also similar to that of distillation, as both methodologies essentially use training techniques to speed up the inference process). Examples of work that probably should be cited and briefly discussed here include but not limited to [2,3,4,5,6,7,16]. More essentially, the authors probably also need to compare the distillation procedure proposed in this work with that of [16] (Or maybe consider including [16] as one of the baselines).

Also, compared to existing work on accelerating the inference of discrete diffusion models like [8], the scale of the numerical experiment on image generation included in this manuscript might be a bit small. Specifically, the authors are encouraged to test the proposed distillation method on large-scale images like FFHQ [9] or ImageNet [10] to better justify its validity.

---

> ### Author Rebuttal · Authors · 2025-07-31
>
> Thanks for your useful comments and questions. Our responses to the main concerns are given as follows. All citations refer to the reference list provided by the reviewer.
>
> (**Examples of work that probably should be cited and briefly discussed here include but not limited to [2,3,4,5,6,7,16].** )
>
> Thanks for the comment. We will add a paragraph discussing the mentioned papers in the revised manuscript, and we can consider citing others if the reviewer has additional suggestions. The paragraph to be added is as follows:
>
> The distillation of continuous diffusion models is a rapidly advancing field. A prominent direction is related to the consistency model [1], which aims to learn a function that maps any point on an ODE trajectory to its origin, enabling one-step or few-step generation. This paradigm has been extended to multi-step variants [2, 3] for improved performance. Other significant works focus on directly matching student and teacher distributions, such as distilling guided diffusion models [4], proposing simplified and faster matching objectives [5], recursively distilling a deterministic diffusion sampler into a new model [6], or concentrating on one-step distillation [7]. While these methods are highly effective for continuous models, they usually rely on continuous paths in the sense that the sampling process of each step is differentiable. Our work diverges by proposing a distillation framework specifically for the discrete diffusion model, which does not assume such a continuous path, and addressing a different set of challenges like the non-differentiability of the outputs. A recent work for Di[M]O [16] also involves distilling discrete diffusion models. It distills a multi-step masked diffusion model into a one-step generator. This is achieved by training a new student model from scratch, using a sophisticated proxy objective that involves creating "pseudo-intermediate states" and training an auxiliary model to match conditional output distributions. Our approaches are significantly different with Di[M]O in both goal and mechanism. Similarly to LD3 and S4S, we focus on a few-step sampler distillation. We tackle the challenge of non-differentiability of sampling from categorical distributions, and we enhance an existing sampler rather than replacing the model, which avoids the complexity of training a new generator and an auxiliary model.
>
> (**The authors probably also need to compare the distillation procedure proposed in this work with that of [16].**)
>
> Thank you for highlighting this work. As mentioned above, we will add a detailed comparison with Di[M]O [16] in the Related Work Section of our paper to clearly position our contribution.
>
> While both Di[M]O and our LSD+ method employ distillation for discrete diffusion models, a direct empirical comparison of the outcomes is not straightforward due to several significant differences between the two methods. Di[M]O trains a new student model from scratch to achieve one-step generation, primarily for image generation tasks. In contrast, our method is a few-step sampler distillation process that trains scalar-form coefficients and time steps for an existing sampler, avoiding the need to train a new model, and is evaluated mainly on text generation tasks. Given these differences in objective (one-step vs. few-step), mechanism (model training vs. sampler training), and application domain (image vs. text), currently, we do not include a comparison with Di[M]O.
>
> Additionally, we note that the work for Di[M]O is publicly available on arXiv on March 19th, 2025. In this context, we would like to politely draw attention to NeurIPS policy, which states: “Papers appearing online after March 1st, 2025 are generally considered concurrent to NeurIPS submissions.  Authors are not expected to compare to those”. (https://neurips.cc/Conferences/2025/PaperInformation/NeurIPS-FAQ)
>
> (**The authors are encouraged to test the proposed distillation method on large-scale images like FFHQ [9] or ImageNet [10] to better justify its validity.**)
>
> Thank you for your suggestion. We conduct experiments on ImageNet (256x256) using the  MaskGIT architecture as the backbone, incorporating the recently proposed advanced Halton sampler (https://arxiv.org/abs/2503.17076) as the sampling method. The results are reported in Table A1, which demonstrates that our LSD+ method yields improved generation performance as measured by the FID metric.
>
> **Table A1: Comparisons on ImageNet 256x256 (in terms of FID$\downarrow$).**
> |Sampler\NFE|4|8|16|32|
> |:---|:--:|:--:|:--:|:--:|
> |Halton|14.16|10.15|8.89|6.92|
> |LSD+-Halton|12.78|8.66|7.17|6.32|
>
> (**Whether it would be possible to provide theoretical justification for the distillation method proposed in this paper by referring to exist work on the theoretical analysis of discrete diffusion models like [8,13,14,15]?**)
>
> Thank you for this question. We agree that providing theoretical justification for our distillation methods would strengthen the work. However, our current focus is primarily empirical: We aim to develop practical and effective techniques for accelerating discrete diffusion models. Given the time constraints of the short rebuttal period, we are unable to establish rigorous theoretical guarantees for our distillation methods at this stage, and we plan to address this with more thorough theoretical analysis in future work.
>
> That said, we would like to offer an intuitive explanation for the effectiveness of our method, which may help lay the groundwork for subsequent theoretical framing. Specifically, the final discrepancy between the outputs of the student and teacher samplers stems from the accumulation of small local errors made at each step. Each of these local errors, in turn, is tied to the specific score predicted by the model at that step. Our training objective directly enforces alignment between the score predictions of the student sampler and those of the teacher sampler at every step. By implicitly correcting these small local errors throughout the process, our approach guides the student sampler to produce final outputs that closely match the high-quality results of the teacher sampler.
>
> (**For Algorithm 3 in Appendix D.2, it seems that some ambiguity exists.**)
>
> Thank you for your careful reading and for bringing this to our attention. We will revise Algorithm 3 to correct the typos therein. The parameters $\tau_k$'s will be updated accordingly after the $\kappa_k$'s get updated. The update rules for $\kappa_k$'s and $\tau_k$'s are as follows:
> $$\kappa\_k \gets \kappa\_k - \eta \nabla\_{\kappa\_k} \tilde{L}\_k$$
> $$\tau\_k = t\_0 - \sum_{l=1}^{k} \kappa\_l$$
>
> (**Whether it would be fair to fix the number of evaluations (NFEs) and compare the performance since the distillation step take nontrivial amount of time?**)
>
> Thank you for the comment and question. Our primary focus is on accelerating the inference stage, where a one-time and offline training cost is traded for significant, repeated latency gains. The training-stage cost is relatively low. For example, under the experimental setup detailed in the Experiment Section of our main paper, distilling a 1024-step teacher sampler to a 16-step student sampler using the SEDD-small backbone takes approximately 4 minutes on a single NVIDIA RTX4090 GPU. By contrast, inference time is substantially higher. Specifically, with a batch size of 1, evaluating perplexity via the generation of 1024 unconditional text samples requires around 300 minutes of inference time on the same GPU. Given that the one-time, offline training cost is orders of magnitude smaller than even a single moderately sized evaluation, we omit training time for our distillation methods and instead focus on reporting performance metrics with fixed NFEs.
>
> This paradigm is not unique to our work. For example, the closely related acceleration method JYS also compares performance under fixed NFEs, despite involving a one-time training cost to learn time schedules.
>
> Regarding high-order solvers, we have included direct comparisons against the $\theta$-RK2 and $\theta$-Trapezoidal methods from [8] in Table 3 of the main paper. However, since the code for [8] is not yet open-sourced, a direct comparison of running times is currently not feasible.

---

> ### Comment · Reviewer_9x6X · 2025-08-05
> **Response to the authors**
>
> The reviewer would like to thank the authors for the detailed response, most of which the reviewer is satisfied with. The authors are strongly encouraged to include all points mentioned in the reviews and rebuttals below, such as literature review on distillation for continuous/discrete diffusion models, comparison with concurrent work like Di[M]O, as well as the additional experimental results on large-scale images like ImageNet.
>
> Moreover, the authors might also consider including examples of generated images from CIFAR/ImageNet in the appendix of the manuscript. Given that theoretical analysis seems to be a primary limitation of the manuscript, the authors should probably consider citing the articles [13,14,15] listed in the references above along with [16] to briefly discuss how one may theoretically analyze the distillation methodology proposed here for discrete diffusion models.
>
> The reviewer will finalize the rating soon.
>
> Added references:
>
> [16] Chen, Hongrui, and Lexing Ying. "Convergence analysis of discrete diffusion model: Exact implementation through uniformization." arXiv preprint arXiv:2402.08095 (2024).

---

> > ### Author Response · Authors · 2025-08-06
> >
> > Thank you very much for your positive feedback and constructive suggestions. We are pleased to learn that our rebuttal has addressed most of your concerns.
> >
> > We will ensure that all the points you mentioned, including the literature review on distillation for continuous/discrete diffusion models, comparison with concurrent work like Di[M]O, as well as the additional experimental results on large-scale images like ImageNet, are incorporated into the revised manuscript.
> >
> > We will also include examples of generated images from CIFAR/ImageNet in the appendix of the revised version. Furthermore, we appreciate your pointing out those four valuable theoretical works; in the revised manuscript, we will ensure these are appropriately cited when discussing the theoretical analysis of our proposed distillation methodology for discrete diffusion models.
> >
> > We sincerely thank you for your valuable comments and suggestions, which will help strengthen our paper.

---

### Comment · Area_Chair_x3qg · 2025-08-05

Dear Reviewers, this is a gentle reminder to read the authors' rebuttal and the other reviews carefully. You are encouraged to post your initial response and engage in the discussion with the authors. This author-reviewer discussion period ends on August 6th AoE. Thank you.

---

> ### Comment · Area_Chair_x3qg · 2025-08-05
>
> Dear Reviewers, the author-reviewer discussion period has been extended to August 8th AoE. Please use this extra time to read the author's rebuttal and engage in discussion with the authors. Thank you.

---

> > ### Author Response · Authors · 2025-08-06
> >
> > Dear AC,
> >
> > Thank you very much for facilitating the discussion period and for your timely reminders. We sincerely appreciate the effort and time you have invested in ensuring a thorough review process.
> >
> > Best regards, Authors

---

### Note · Authors · 2025-08-14

Dear AC,

Thank you for overseeing and facilitating the discussion period. We would like to briefly summarize the outcomes of our recent efforts as the deadline for final decisions approaches.

We are pleased that our detailed rebuttal has addressed the majority of the concerns initially raised by the reviewers, leading to their positive feedback and responses. We are particularly encouraged by the feedback shared by **Reviewer 61AF, who explicitly noted: “I would increase by final rating after discussion period.”**

Per the records in the OpenReview system, we note that Reviewer 61AF may not yet have submitted their final rating. We would be deeply grateful if the reviewer could take a moment to update their rating at their earliest convenience.

As the deadline for providing final ratings approaches, we also respectfully wish to bring this positive comment to your attention—and, by extension, to the reviewers'—for consideration. We recognize that amid the busyness of consolidating final reviews, such details may occasionally be overlooked, and we hope Reviewer 61AF's earlier expression of support (noting they would adjust the rating positively following the discussion period) is not inadvertently missed.

We would like to take this opportunity to express our sincere gratitude for the thorough, constructive, and thoughtful review process. Each round of feedback has strengthened our work, and we deeply appreciate the time and care you and the reviewers have invested.

Thank you again for your attention to this matter, and for your ongoing guidance.

Best regards, Authors

---

### Decision · Program_Chairs · 2025-09-17

**Decision:**

Accept (spotlight)

**Comment:**

This paper introduces a novel method, Learnable Sampler Distillation (LSD) and its extension LSD+, to accelerate the sampling process of discrete diffusion models (DDMs). The core idea is to train a lightweight "student" sampler to mimic the behavior of a high-fidelity, multi-step "teacher" sampler. The approach achieves this by optimizing a small number of learnable parameters: per-step coefficients that modify the score predictions and, for LSD+, an adaptive time schedule. The paper demonstrates that LSD/LSD+  improves the trade-off between generation speed and quality across multiple tasks, including text and image generation, and on various model architectures.

The proposed method is intuitive, and its empirical results are impressive. As noted by Reviewers 9x6X, 61AF, and VCFx, the paper shows performance gains over strong baselines, with fewer sampling steps (NFEs). The method's low computational overhead for training and its compatibility with various backbones and other acceleration techniques (e.g., remasking, parallel decoding) make it valuable to the community.

The primary weakness, highlighted by Reviewers 9x6X, ACPU, and odGL, is the lack of a strong theoretical foundation. The paper's core objective, which aligns scores at each step, is presented as an effective heuristic without a formal analysis of why this approach guarantees a better final sample distribution. A related point raised by Reviewer ACPU is the underexplored design choice of using scalar coefficients instead of more expressive forms, such as vector-based coefficients. Although the authors provide an intuitive explanation, the absence of rigorous theoretical guarantees remains a notable limitation.

I recommend this paper for spotlight acceptance because it offers an exceptionally strong and practical solution to a critical problem in the DDM space. The empirical results are not only compelling but also thoroughly validated through the rebuttal process. The authors' commitment to rigorous empirical evaluation, as demonstrated by the extensive new experiments, elevates the paper to "spotlight" status, provided that all promised improvements are fully integrated into the camera-ready version.


The author-reviewer discussion was productive, with the authors effectively addressing nearly all concerns. Reviewers 9x6X and ACPU initially raised questions about the paper's limited scope and scalability. The authors responded by providing extensive new results on larger datasets, which successfully demonstrated the method's robustness and scalability. Reviewer 61AF inquired about integrating the method with other techniques and solvers, and the authors provided new data showing successful integration with predictor-corrector solvers, remasking, and parallel decoding. The lack of a theoretical foundation was a key concern for Reviewers 9x6X, ACPU, and odGL. While the authors could not provide a formal proof during the rebuttal period, their intuitive explanation for the method's effectiveness was well-received and helped all reviewers feel confident in the empirical results. Reviewer VCFx raised concerns about the small scale of the experiments and the robustness to hyperparameters; the authors provided a new ablation study on the Hamming distance threshold that addressed those concerns. Most of the reviewers either maintained their positive score or raised their rating, leading to a  consensus for acceptance.